# Building on Efficient Foundations: Effectively Training LLMs with Structured Feedforward Layers

**Xiuying Wei**
xiuying.wei@epfl.ch
CLAIRE, EPFL

**Skander Moalla**
skander.moalla@epfl.ch
CLAIRE, EPFL

**Razvan Pascanu**
razp@google.com
Google DeepMind

**Caglar Gulcehre**
caglar.gulcehre@epfl.ch
CLAIRE, EPFL

## Abstract

State-of-the-art results in large language models (LLMs) often rely on scale, which becomes computationally expensive. This has sparked a research agenda to reduce these models' parameter counts and computational costs without significantly impacting their performance. Our study focuses on transformer-based LLMs, specifically targeting the computationally intensive feedforward networks (FFNs), which are less studied than attention blocks. We consider three structured linear parameterizations of the FFN using efficient low-rank and block-diagonal matrices. In contrast to many previous works that examined these approximations, our study i) explores these structures from a training-from-scratch perspective, ii) scales up to 1.3B parameters, and iii) is conducted within recent Transformer-based LLMs rather than convolutional architectures. We demonstrate that these structures can lead to actual computational gains in various scenarios, including online decoding when using a pre-merge technique. Additionally, we propose a novel training regime, called *self-guided training*, aimed at improving the poor training dynamics that these approximations exhibit when used from initialization. Interestingly, the scaling performance of structured matrices is explored, revealing steeper curves in scaling training FLOPs, along with a favorable scaling trend in the overtraining regime. Specifically, we show that wide and structured networks can utilize training FLOPs more efficiently, with fewer parameters and lower loss than dense models at their optimal trade-off. Our code is available at https://github.com/CLAIRE-Labo/StructuredFFN/tree/main.

Table 1: **Better training FLOPs utilization of the wide and structured Networks**: we compare dense Transformers trained according to their optimal scaling law [1], efficient Transformers (GQA) [2] with high throughput, and our wide and structured networks using `LowRank` parameterization in the FFN module and reduced attention heads, under the same training FLOPs. TP (throughput) refers to the maximum throughput measured over a generation length of 256.

| Method | #Param | Training FLOPs | PPL | TP (token/s) |
|---|---|---|---|---|
| Transformer-m | 335M | 1.55e+19 | 18.29 | 30229 |
| Transformer-m (GQA) | 335M | 1.55e+19 | 18.23 | 84202 |
| **Wide and Structured** | **219M** | 1.55e+19 | **17.89** | **91147 (8% ↑)** |
| Transformer-l | 729M | 7.03e+19 | 14.29 | 23351 |
| Transformer-l (GQA) | 729M | 7.03e+19 | 14.40 | 64737 |
| **Wide and Structured** | **464M** | 7.03e+19 | **14.27** | **75930 (17% ↑)** |

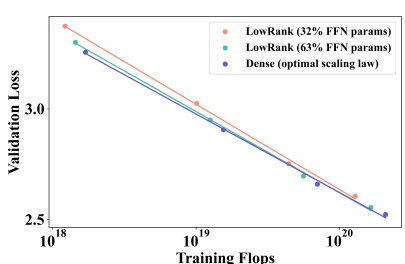

Figure 1: **Steeper scaling curves of `LowRank` with 63% or 32% FFN parameters**. For more results, see Sec. 4.2.

38th Conference on Neural Information Processing Systems (NeurIPS 2024).

# 1  Introduction

Transformer language models [3] have gained significant attention for their performance and scalability. These models have grown from hundreds of millions of parameters [4] to hundreds of billions [5–7], increasing the need for efficient training and inference techniques. While much research focuses on attention, *feed forward network*s (FFNs) account for over 60% of the model's parameters and FLOPs, significantly impacting latency.[1] Recent large-scale models [8, 9] further increase the FFN size, leading them to dominate the cost of the model compared to the attention layer.

Structured linear transformations, such as low-rank or block-diagonal matrices, are important paradigms for reducing the computational cost of feedforward layers. However, they have not yet been thoroughly explored at a sufficient scale to reduce pre-training costs and latency of the inference phase in modern LLM architectures, where the main focus so far has been on improving the efficiency of the self-attention mechanism.

In this work, we investigate structured matrices for FFN blocks from the train-from-scratch aspect, first identifying their efficiency and optimization challenges and then presenting experimental results, analyzing and characterizing the behavior of models trained with structured matrices, and comparing their results. We consider three efficient linear parametrizations: `LowRank`, `BlockShuffle` (comprising two block-diagonal matrices), and `BlockDense` (a combination of dense and block-diagonal matrices). First, while they have demonstrated materialized computational gains, they face challenges in the practical online decoding scenario of LLM, which may process only limited input tokens at one time, leading to under-utilization of computing resources and decreased efficiency due to the additional linear projection. We address this with a pre-merge technique that restores efficiency to the original dense parametrization. Second, we observe that these parameterizations of the FFN blocks are harder to train than standard linear layers, often exhibiting poorer training dynamics like loss spikes. To counter this, we propose a flexible and fast method we refer to as *self-guided training*. It employs a dense matrix as a residual component during the initial training phase, steering the training process away from suboptimal starting points gradually.

We conduct our experiments at scale on Transformers ranging from 110M to 1.3B parameters by replacing the traditional heavy FFN with structured matrices. Our experiments first show the scaling behavior of these structured linear parameterizations and then illustrate how our proposed methods address their general efficiency and optimization challenges. First, we examine scaling performance from the perspectives of training compute and model size, highlighting the potential of structured matrices. By controlling for the same training FLOPs, we find that structured FFNs show steeper loss scaling curves than traditional Transformers at optimal trade-offs (See Fig. 4 and Fig. 1). Specifically, as seen in Table 1, our wide and structured networks use training FLOPs more efficiently, needing fewer parameters (464M vs. 729M) and achieving a 17% throughput boost on the -l scale, while still maintaining slightly better perplexity compared to the efficient Transformer [2]. Beyond training compute scaling, we also scale model size in the overtraining regime, with Fig. 5 showing favorable scaling trends for our wide and structured models. Second, our results on efficiency show that structured FFNs, with only 32% of the FFN parameters, can boost the training speed of the 1.3B model by $1.35\times$. Furthermore, self-guided training enhances the performance of all three structured matrices (e.g., reducing the perplexity gap of `LowRank` to about 0.4) without affecting the inference time speed-up.

As the first work to explore structured matrices at the scale of recent LLMs, we hope our findings and results will shed new light on the study of efficient NLP architectures. Our contributions can be categorized into the following three aspects:

1. We investigate three types of structured matrices in Transformer pretraining and demonstrate their favorable scaling behavior compared to dense models. This is revealed through the study of scaling laws for training FLOPs, as well as model size scaling in the overtraining regime, showing that wide and structured networks can be strong candidates for architecture design.
2. We conduct an efficiency study of these structured matrices across various scenarios. We propose a pre-merge technique to maintain speed in a specific case and show the effective speed-up performance of structured matrices in other scenarios.
3. We identify optimization challenges in structured matrices and introduce a method called self-guided training, which efficiently improves training dynamics and boosts the final performance for all three types of structured matrices.

---

[1]For example, we find that it composes 54% of total latency in a 1.3B model.

## 2 Method

Multiple techniques have been proposed to approximate linear projections from sparsity to low-rank approximations. We provide an in-depth discussion of existing literature on approximating linear layers in section 3, that better contextualizes our work. We focus on structured approximations of linear projections $g(\boldsymbol{x}) = \boldsymbol{W}\boldsymbol{x}$ that have the form $g(\boldsymbol{x}) = \boldsymbol{U}(\boldsymbol{V}\boldsymbol{x})$, where $\boldsymbol{U}$ and $\boldsymbol{V}$ are structured matrices, e.g. low-rank or block-diagonal. We opt for this particular form because it allows us to readily exploit the computational gains on existing hardware using existing libraries with minimal alteration. Such approximations have been previously studied in different contexts. Our contributions are exploring them (i) to approximate FFN layers of transformers, (ii) when applied from initialization, (iii) testing them at up to 1.3B scale, investigating their general bottlenecks and providing scaling analyses.

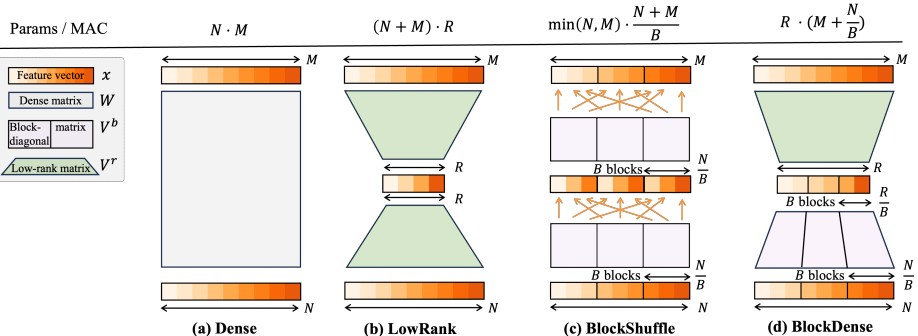

Figure 2: **Structured linear parametrization:** We show the structured linear parametrization with input dim. of $N$ and output dim. of $M$. a) The traditional dense linear parametrization. b) `LowRank` parametrization with a bottleneck of size $R$ where $R$ is less than $M$ and $N$. c) `BlockShuffle` with two block-diagonal matrices with blocks of size $B$ interleaved with a shuffle operations that mixes information from different blocks similar to ShuffleNet. d) `BlockDense` with the first matrix as a block-diagonal and the second a low-rank or dense matrix.

### 2.1 Structured linear parametrization

We explore three structured matrices to approximate the standard linear layer $\boldsymbol{W}\boldsymbol{x}$, maintaining its input dimension $N$ and output dimension $M$ of weight $\boldsymbol{W}$.

`LowRank`  Low-rank matrices have been widely used to decompose pre-trained weights for downstream compression [10] or to construct adapters for efficient fine-tuning [11]. Researchers [12] suggest that dense layers tend to naturally converge to low-rank solutions during training, making this approximation ideal. Inspired by this, we explore low-rank matrices as alternatives to traditional linear layers, imposing this structure from initialization and investigating it during pre-training.

Formally, the low-rank approximation of a linear layer is given as $\boldsymbol{W}\boldsymbol{x} \approx \boldsymbol{U}^r(\boldsymbol{V}^r\boldsymbol{x})$ where $\boldsymbol{U}^r \in \mathbb{R}^{M \times R}$, $\boldsymbol{V}^r \in \mathbb{R}^{R \times N}$ and $R < \min(M, N)$. Note that we use the superscript $^r$ to indicate that these matrices are used to create a low-rank approximation by projecting to or from a low-dimensional code, a notation that would become useful later on to distinguish such components from block-diagonal ones. The parameter count and MAC (Multiply-Accumulate Operations) decrease from $M \cdot N$ to $(M + N) \cdot R$.

`BlockShuffle`  Dao et al. [13] proposes using the Monarch decomposition of FFT, replacing the linear layer with interleaved block-diagonal and permutation matrices. An alternative motivation for such a structure can be derived from efficient convolutional designs of ShuffleNet [14] and separable convolution [15]. For simplicity, we explore the form introduced by ShuffleNet to linear layers.

The core idea of `BlockShuffle` is to reshape the feature dimension into two dimensions and first use a linear projection that mixes along one of these fictive dimensions, followed by a linear projection that mixes along the other. More precisely, we first reshape the input features $\boldsymbol{x} \in \mathbb{R}^N$ into $B$ blocks and apply the non-tied weight of $\frac{N}{B} \times \frac{N}{B}$ to each block, then flatten the intermediate feature. To achieve global perception, we regroup elements from different blocks into $B$ new blocks and apply the same transformation for each block again.

Technically, we can express the per-block transformation using block-diagonal matrices and formulate the above process as $\boldsymbol{W}\boldsymbol{x} \approx f^{-1}(\boldsymbol{U}^b f(\boldsymbol{V}^b \boldsymbol{x}))$, where block-diagonal matrices $\boldsymbol{V}^b$ and $\boldsymbol{U}^b$ has $B$ blocks with shapes $\frac{\min(N,M)}{B} \times \frac{N}{B}$ and $\frac{M}{B} \times \frac{\min(N,M)}{B}$ per-block. As shown in Fig. 2, the shuffle function $f(\cdot)$ enables global feature mixing by cycling different blocks and can be implemented by simply transposing and reshaping inner features. The inverse function $f^{-1}(\cdot)$ permutes the outputs back to their original order.

By separating features into two dimensions, only a few elements of the features will be processed each time. The parameter count and MAC are reduced from $M \cdot N$ to $\min(N,M) \cdot \frac{(M+N)}{B}$, where $B$ acts as a trade-off of accuracy and efficiency.

`BlockDense`   The previous parametrization incorporates additional shuffle operations, which can be slow on the device. We propose a natural intermediate candidate between `LowRank` and `BlockShuffle`, combining the block-diagonal projection $\boldsymbol{V}^b$ with a dense or low-rank projection $\boldsymbol{U}^r$. Thus, we can mix the features between blocks without permuting the inner features. The formula is defined as $\boldsymbol{W}\boldsymbol{x} \approx \boldsymbol{U}^r(\boldsymbol{V}^b \boldsymbol{x})$, where $\boldsymbol{V}^b$ is the block-diagonal matrix with $b$ blocks in shape $\frac{R}{B} \times \frac{N}{B}$, and $\boldsymbol{U}^r \in \mathbb{R}^{M \times R}$. Technically, the second projection does not need to be a low-rank approximation because $R$ can be larger than $M$. Nevertheless, in practice, we chose $R < M$ to limit the search space of this work, and thus use the superscript $r$ for the second matrix. The parameters of this method are determined by two variables $B$ and $R$, cutting the original burden from $M \cdot N$ to $R \cdot (M + \frac{N}{B})$. Note that `BlockDense` can recover the `LowRank` approximation if we set $B = 1$ and $R < \min(M,N)$.

**Remark**   We limit our exploration of the efficient linear parametrizations within the FFN blocks. These typically have $8 \cdot H^2$ parameters and MAC, where $H$ standards for hidden state dimension and $4 \cdot H$ as the intermediate hidden size of FFN. In contrast, the proposed parametrizations have:

$$\texttt{LowRank:}\, 10 \cdot H \cdot R \qquad \texttt{BlockShuffle:}\, 10 \cdot \frac{H}{B} \qquad \texttt{BlockDense:}\, 5 \cdot H \cdot R \cdot (1 + \frac{1}{B})$$

Although `BlockDense` is introduced as a new parameterization, the aim of this paper is not to claim it as the best candidate, but rather to investigate some general properties of structured matrices from efficiency, optimization, and scaling perspectives. Given the favorable efficiency and loss performance of `BlockDense`, it is included alongside `LowRank` and `BlockShuffle` here to cover a broader range of potential parameterizations.

## 2.2   Maintaining efficiency during online decoding

**Parallelism-bound FFN** With reduced FLOPs and parameters, our proposed linear parametrization can accelerate the model for compute-bound and memory-bound scenarios [16], usually during training, prefilling, and decoding with a relatively big batch size. However, for online decoding with a very small batch size and sequence length of 1, a practical scenario for LLM, both FFN and structured FFN can become parallelism-bound [17] with poor utilization of the GPU resources, especially on powerful devices like A100. Because each linear projection suffers from parallelism-bound, efficient linear parametrization may lead to worse latency performance due to doubling the number of linear projections. We propose a *pre-merge technique* to mitigate the issue.

**Pre-merge technique** Taking advantage of the fact that these parametrizations do not have non-linearity, we propose to combine the structured matrices into a single dense layer and keep both the structured and the dense one for online decoding. Then, we can dynamically decide which parametrization to use based on the current batch size and setting. Fig. 7 analyzes using structured or dense forms for different batch and model sizes, allowing us to decide when to use the pre-merged linear projection.

## 2.3   Addressing the optimization challenge

Using the efficient parametrization from initialization can suffer from optimization difficulty because the deep linear parametrization introduces additional symmetries[2], which is a source of proliferation

---

[2]Symmetries arise because such a factorization $\boldsymbol{U}\boldsymbol{V}$ is not unique; for any invertible matrix $\boldsymbol{C}$ of size $R \times R$ we have $\boldsymbol{U}\boldsymbol{V} = \boldsymbol{U}\boldsymbol{C}\boldsymbol{C}^{-1}\boldsymbol{V} = (\boldsymbol{U}\boldsymbol{C})(\boldsymbol{C}^{-1}\boldsymbol{V}) = \tilde{\boldsymbol{U}}\tilde{\boldsymbol{V}}$. `BlockShuffle` can also be included by fusing the shuffle operation into block-diagonal matrices.

of saddle points and generally less smooth loss function as pointed out in [18, 19]. We hypothesize that this makes poorer learning dynamics of the structured parametrization. Empirically, we found that the deep linear form $U(Vx)$ is more difficult to train than the standard linear layer. For example, in Fig. 3, it can suffer from training instability and loss spikes with a large learning rate, while also converging much more slowly than the dense form with a small learning rate. We further elaborate on this, highlighting how the inconsistency of gradient updates between the structured parametrization and original linear projection affect learning dynamics in Appendix B.2.

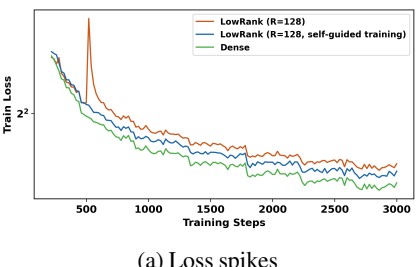 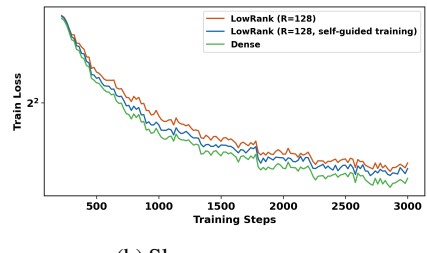

(a) Loss spikes                                         (b) Slow convergence

Figure 3: **Poor training dynamics:** Training dynamics of `LowRank` with rank of 128 under different training configurations. Curves correspond to a 4-layer Transformer with a model width of 768 on WikiText-103. We apply self-guided training in the first half of training. Refer to Sec. B.1 for more training dynamics visualizations of the other two structured parameterizations.

**Self-guided training** Addressing the poor training dynamics by carefully tuning the learning rate schedule and gradient clipping coefficient might be possible, but it is costly and may switch between slow convergence and training instability. We propose a less costly and simple approach that can be used with minimal re-tuning of hyperparameters.

To motivate our proposal, we start by finding that the updates on $UV$ scales the function of the backpropagated gradients $g$ (see App. B.2), then turn into the typical training dynamics with gradients. An issue that needs to be addressed in the early stage of training is feature specialization when the learning process assigns semantics to the different hidden units of the model, sometimes also phrased as identifying the winning ticket [20]. In this process, certain weights will need to be suppressed, and symmetries in the parametrization must be resolved.

To address this problem, we propose using the dense parametrization $W$ as a crutch to efficiently make decisions about feature specialization and then transfer them to $U$ and $V$ through $g$. To this end, we use the following parametrization

$$o = \alpha \cdot Wx + (1-\alpha) \cdot U(Vx). \tag{1}$$

$o$ is the layer's output, and $\alpha$ decays following a cosine scheduler. As a residual component, learning $W$ is unaffected by the additional saddles and pathologies introduced by the structured parametrization, allowing units to specialize. This *guides* the training of $U$ and $V$, which are forced slowly to take over by providing the hidden units semantics learned by $W$. This approach also relates to homotopy methods such as *simulated annealing*, where a hard optimization problem is transformed into an *easier* to optimize form with certain desired properties, gradually transforming the problem to its original form. Here, we consider the easier optimization problem is to train with dense matrices. By decreasing alpha from 1 to 0, we transform the easier-to-optimize loss into the original parametrization we want to use.

Furthermore, we initialize $W_0 = U_0 V_0$, making the observation that by using this initialization for the dense residual branch, we can easily start the *guided training* at any stage of learning (e.g., fine-tuning) without affecting the model behavior. We refer to this as self-guiding training, as the better learning dynamics from $W$, which initially is just $UV$, guide the learning of $U$ and $V$ through the backpropagated gradients $g$.

Guided by the dense weights, which do not have the symmetry problem, it becomes much easier for the structured matrices to learn a good representation. From Fig. 3, it can be observed that the self-guided training prevents training spikes and fastens the convergence process. Notably, it benefits all three structured matrices with improved training dynamics illustrated in Sec. B.1 and better final performance shown in Sec. 4.4.

**Reducing the computational cost of self-guided training:** Note that while $\alpha > 0$, we need to perform forward and backward passes through the dense version of the weight matrix $\boldsymbol{W}$, which could be expensive. To address this issue, we consider the following stochastic version of the above formula, which allows us to control how often we need to use the dense residual branch:

$$\boldsymbol{o} = \begin{cases} \alpha \cdot \boldsymbol{W}\boldsymbol{x} + (1-\alpha) \cdot \boldsymbol{U}(\boldsymbol{V}\boldsymbol{x}), & p < \alpha \\ \boldsymbol{U}(\boldsymbol{V}\boldsymbol{x}), & p \geq \alpha. \end{cases} \tag{2}$$

In our practice, $p$ is a random variable sampled independently from a uniform distribution over 0 to 1 in each training forward pass. With $\alpha$ following a cosine annealing schedule, this softer version Eq. (2) reduces the expected computation to half of Eq. (1). For example, using our method for half the training time increases the FLOPs by only 25% of the original FFN. This has a negligible impact on accuracy. More ablation studies of this technique are presented in Sec. B.3.

# 3 Related work

**Efficient techniques for training LLMs** Recent advancements in attention mechanisms have significant improvements in the efficiency of attention [21–23, 2, 24–29], and the focus has shifted towards improving FFNs in LLMs, which contributes to at least half of the training time. Dynamic architectures such as mixtures of experts [30–32], or optimizers with faster convergence [33, 34] have been popular in improving training efficiency. Moreover, Xia et al. [35] employs structured pruning with learned sparsity masks and a dedicated data-loading policy to reduce the training budget.

There has been a recent focus on parameter-efficient fine-tuning methods like LoRA [11] and structured approximation of the linear layers (see, [10]). LoRA uses the low-rank approximation to reduce trainable parameters during the finetuning phase, whereas Sharma et al. [10] selectively applies low-rank decomposition to well-trained weights. While these methods used low-rank approximation of the weights, they did not focus on pre-training.

**Structured matrices in deep learning** Researchers use structured matrices in the form of dense matrices with shared parameters, like Circulant and Toeplitz [36][3], and structured matrices, such as low-rank and diagonal, to reduce parameters and FLOPs while optimizing CUDA kernel use. Low-rank matrices, initially used in convolutional networks [37], have shown high efficiency in training [38], achieving up to a 2.9× speed-up with similar performance. Some studies [39, 40] adapt the rank during training and suggest regularizers to maintain SVD decomposition for better accuracy. Khodak et al. [41] propose spectral initialization and aligning weight decay of matrix products with standard linear layers. However, these studies mainly focus on ResNets [42] rather than recent LLMs. There have been other studies that aim to improve the expressiveness of structured matrices. For instance, Moczulski et al. [43] uses interleaved diagonal and Fourier transforms, while Dao et al. [44] proposes butterfly parametrization for various transformations. These approaches often lack efficiency due to additional layers or irregular sparsity patterns. Dao et al. [13] simplified butterfly matrices to block-diagonal ones, achieving a 2× speed-up on WikiText-103 language modeling tasks. In this work, for accuracy and efficiency, we explored low-rank factorization of weight matrices with reduced bottleneck dimension and block-diagonal matrices to reduce parameters in our LLM training studies.

# 4 Experiments

In our experiments, we empirically analyzed the performance of scaling, efficiency, and self-guided training for structured parameterization in LLMs.

## 4.1 Settings

**Model** We perform the experiments on the standard Transformer architecture [45, 4] equipped with *rotary positional embeddings* [46] and the Llama Tokenizer [47]. Its FFN module is composed of two linear layers and a GeLU activation. Four sizes are considered, including Transformer-s (110M), Transformer-m (335M), Transformer-l (729M), and Transformer-xl (1.3B). For our efficient

---

[3]We ran preliminary experiments using Circulant, Toeplitz matrices, and convolutions to improve the efficiency of FFNs, but our initial results were negative (slow and worse performance) and we did not pursue this direction further.

parameterizations, we only make the FFN module structured in most experiments to simplify our study, as the attention module has been well-studied [2, 23]. We explore two sizes that retain 63% or 32% of the dense FFN parameters by adjusting the rank and number of blocks (e.g., using a rank half the FFN width in `LowRank` reduces the parameters to 63%). In particular, to provide more comparative results with dense models in the scaling study Sec. 4.2, we further design the *wide and structured networks*, where both the attention and FFN modules are made structured using [2] and the structured matrices. This is because allocating more parameters to the FFN compared to the attention module is more favorable, and making them both structured helps maintain the parameter ratio between them. Detailed configurations can be found in Table 10.

For implementation, we take Dao et al. [13]'s implementation for the `BlockShuffle` and carefully manage memory copies for `BlockDense`. In our experiments, we chose $B$ as a common divisor of $M$ and $N$ or $R$. Proper initialization is also investigated in Sec. A.1.

**Training**    We use the RefinedWeb dataset [48] and randomly select 0.5B tokens for validation, reserving the rest for training. All experiments, except for the overtraining experiments on 300B tokens in Fig. 5, are based on the Chinchilla scaling law [1], where tokens are allocated at 20 times the baseline model size. We set hyperparameters such as learning rates and global batch size according to the scaling law studies from recent papers [49, 50]. However, for the 300B token experiments, we found that more advanced hyperparameter settings are necessary. For example, we use betas of [0.9, 0.98]. Additionally, different works tend to use very different learning rates [51, 49, 52, 47] in the overtraining regime. Thus, we follow the scaling law of hyperparameters described in Bi et al. [53] to avoid extensive hyperparameter searches. Other implementations include using A100 80G GPUs, mixed precision (`bfloat16` and `float32`), and adopting fast CUDA kernels like Flash Attention [25] for all experiments. We measure training FLOPs as in Megatron [54], including all matrix multiplications. Additional details are provided in Appendix C.

## 4.2    Scaling analysis

We evaluate the scaling performance of structured linear parameterizations from two perspectives. The first study investigates the scaling law of training-time compute. The second study trains these models with 300B tokens and evaluates their performance on downstream tasks. The results show that our structured matrices can serve as a strong alternative to the dense FFN, utilizing training FLOPs more efficiently (e.g., smaller model and lower loss) and performing better in the overtraining regime.

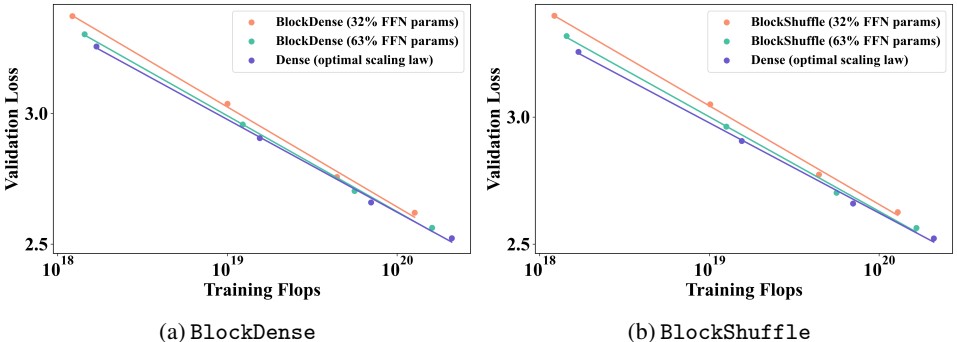

Figure 4: Scaling curves of structured matrices with a linear fit for better illustration. The dense model is trained at its optimal trade-off while we train structured FFNs on the same number of tokens and retain 63% or 32% of the original parameters. 1) Structured matrices have steeper scaling curves with much closer results at larger sizes, showing good scaling behavior. 2) With the same training FLOPs, these curves indicate that structured matrices can have fewer parameters and lower validation loss when the x-axis is further extended.

**Scaling law study: better training FLOPs utilization**    Based on the Chinchilla scaling law, we train four sizes of Transformer models and then use the same amount of tokens to train the structured alternatives. First, we only make the FFN module structured to build the basic understanding, and retain 63% or 32% of the original parameters. In Fig. 4 and Fig. 1, we apply a linear fit to the scaling points for better illustration and show that all three structured matrices have *steeper scaling*

*curves* compared to the dense models, indicating the significant potential of highly structured large models. More importantly, by fixing the training FLOPs, they have fewer parameters and eventually achieve very close or even slightly lower loss (e.g., `LowRank` with 63% parameters). Given their steeper scaling curves, we can also expect noticeably lower loss and fewer parameters for structured parameterizations per FLOP when the x-axis is further extended. Detailed numbers are provided in Table 6 in the appendix, with comparisons among the three structured parameterizations.

Next, the attention module is also structured using GQA [2], resulting in wide and structured networks. This further optimizes the use of training FLOPs, addressing the imbalance caused by structuring only the FFN module, which increases the relative impact of the attention module on the overall architecture. We adopt `LowRank` as an example, as it demonstrates superior performance compared to the other two approaches in our settings, as demonstrated in Table 6 and Fig. 4. To align the training FLOPs, the wide and structured networks are trained on a larger number of tokens. It can be observed in Table 1 that these models achieve lower perplexity while using much fewer parameters. For instance, the parameter count can be reduced from 729M to 464M without compromising perplexity. Additionally, in terms of maximum throughput, ours models achieve an 8% and 17% boost on Transformer-m and Transformer-l, respectively, compared to the fast GQA.

In conclusion, the structured matrices and the wide and structured networks demonstrate great potential in optimizing training FLOP utilization, achieving lower loss with fewer parameters. Additionally, it is important to note that our scaling curves for the structured matrices are not drawn at their optimal training-compute trade-off, while the baseline is.

**Scaling model size: better downstream performance**  To further illustrate the potential of structured matrices, we consider the overtraining regime and use `LowRank` as an example. Specifically, we train four sizes of the dense model on 300B tokens, and build the wide and structured networks upon the design of the dense models by applying `LowRank` to the FFN and reducing the number of attention heads to make the entire network structured. Then, the well-trained models are evaluated on downstream tasks, including PIQA, HellaSwag, Winogrande, and ARC tasks, using `lm-evaluation-harness`[4] with the default prompt. Fig. 5 presents the results, displaying the scaling trend across the four tasks (see detailed numbers and additional tasks in Table 8). The wide and structured models demonstrate comparable or superior performance, particularly at larger sizes, solidifying their benefits over dense architectures.

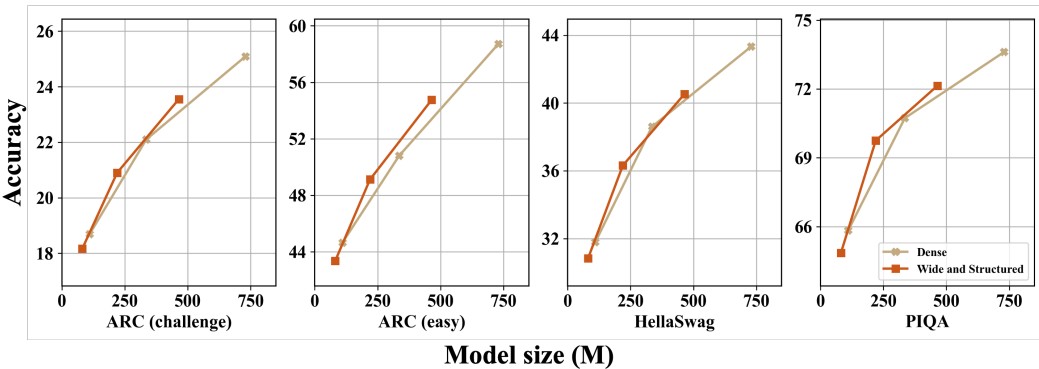

Figure 5: Zero-shot performance on downstream tasks in the overtraining regime. The wide and structured networks are built upon dense ones by applying `LowRank` to the FFN and reducing the number of attention heads to make the entire network structured.

## 4.3  Efficiency study

We investigate the efficiency of structured FFN and consider different numbers of tokens $T$ to discuss different scenarios. Here, $T$ corresponds to the total number of tokens in a batch.

---

[4] `https://github.com/EleutherAI/lm-evaluation-harness`

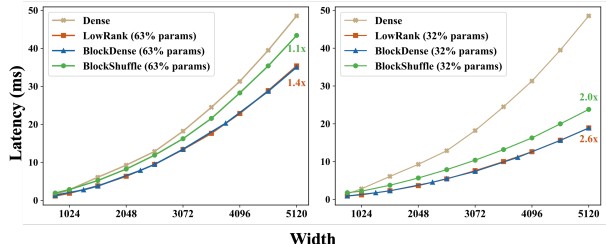

Figure 6: Latency of structured and dense FFNs across different FFN widths. Results are evaluated on 30000 tokens. The intermediate size of the FFN is set to be 4 times the FFN width.

Table 2: Training time of Transformer-xl and structured counterparts with 32% and 63% FFN parameters.

| Model | Params. (M) | Training time (h) | PPL |
|---|---|---|---|
| **Transformer-xl** | 1274 | 352.2 | 12.46 |
| 63% LowRank | 985 | 302.2 | **12.86** |
| 63% BlockDense | 955 | **298.7** | 12.97 |
| 63% BlockShuffle | 985 | 330.6 | 12.98 |
| 32% LowRank | 744 | **260.2** | 13.55 |
| 32% BlockDense | 728 | 261.2 | 13.74 |
| 32% BlockShuffle | 744 | 284.9 | 13.81 |

**Large number of tokens** Using large $T$, the standard linear layers and our efficient structured parametrizations become computation-bound where FLOPs become a latency bottleneck [16]. This setting mainly concerns training, the prefill phase of inference, and extensive offline decoding. In Fig. 6, we evaluate the latency performance of structured and dense FFNs across different FFN widths with 30K tokens. With parameters and FLOPs reduced to 63% or 32%, the lowrank and `BlockDense` achieve a 1.4× or a 2.5× speed-up, respectively. `BlockShuffle` offers modest improvements, with 1.1× and 2.0× speed-ups for the two cases. We also measure the training time of the whole model in Table 2, and observe that `LowRank` with 63% FFN parameters reduces the training time by about 15% with 0.4 increased perplexity, and the one with 32% FFN parameters offers 1.35× whole training speed-up with 1.1 increased perplexity.

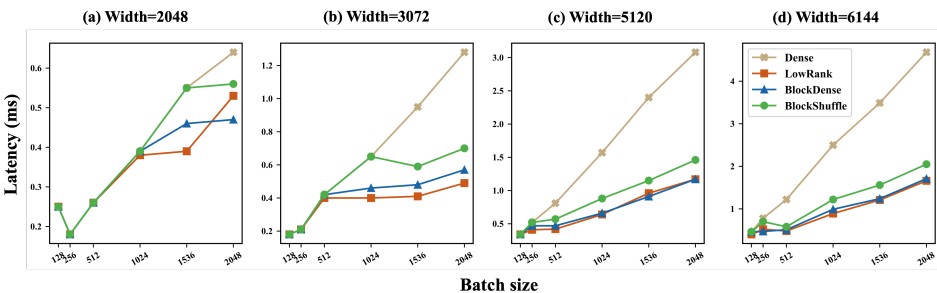

Figure 7: **Latency over different batch size for different widths:** Decoding latency results between dense FFN and structured matrices with 32% FFN parameters across different widths and batch sizes. Note that we have a sequence length of 1 at the decoding phase; thus, $T$ equals batch size.

**Small number of tokens** FFN can be parallelism-bound with small $T$ (e.g., $T = 16$) on the A100 GPUs. Then, when $T$ gets increased, FFN becomes memory-bound and will eventually be computation-bound. Online and offline decoding stages may encounter a small number of tokens when unrolling the model step by step. As discussed earlier, our pre-merge method can alleviate the parallelism-bound issue and maintain the same latency with dense matrices. Fig. 7 shows the latency results for three different widths, varying the batch of tokens to determine when to use efficient alternatives or choose pre-merged dense matrices. For example, with a 2048-width FFN, it is difficult to fully utilize resources on GPU with limited tokens. The performance improves significantly when using width 5120 and 6144, such as speed improvements of 2.63× speed-up of `LowRank` with 32% FFN parameters on $T = 2048$ and 2.81× acceleration of `BlockDense` with 32% parameters on $T = 1536$.

### 4.4 Self-guided training

We apply self-guided training during the first half of training to demonstrate its effectiveness. As shown in Table 3 and Table 9, our method consistently reduces loss across all efficient parametrizations, improving the perplexity by 1.2 for Transformer-s and 0.8 for Transformer-m. Then, to enable a straightforward comparison under the same training FLOPs, we adjust the training steps for self-guided training and repeat those tokens at the end to ensure they're fully learned by structured matrices. As can be seen in Table 9 and Fig. 14a, Fig. 14b, Fig. 8, this reduces the perplexity

gap for Transformer-xl from 1.0, 1.2, and 1.3 to 0.4, 0.5, and 0.6 for `LowRank`, `BlockDense`, and `BlockShuffle`, respectively, under the same training FLOPs and can still enjoy 32% model FLOPs, which can bring about $2.6\times$ inference speed-up. Additionally, we compare our method with another advanced baseline that trains structured matrices with more tokens, showing that the self-guided training can achieve comparable or superior results even with the same number of tokens.

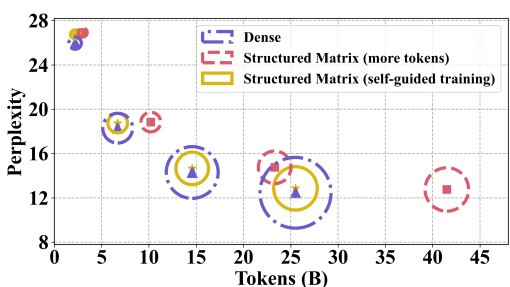

Table 3: Performance of the three structured parameterizations when applying self-guided training♣ in the first half of training. This increases 25% FFN training FLOPs. For more comparisons, please refer to Sec. B.3.

| Architecture | FFN | Training FLOPs | PPL |
|---|---|---|---|
| **Transformer-m** | 201M | 1.55e+19 | 18.29 |
| `LowRank` | 69M | 1.01e+19 | 20.60 |
| `LowRank`♣ | | 1.21e+19 | **19.90** |
| `BlockDense` | 65M | 1.00e+19 | 20.85 |
| `BlockDense`♣ | | 1.19e+19 | **20.10** |
| `BlockShuffle` | 69M | 1.01e+19 | 21.12 |
| `BlockShuffle`♣ | | 1.21e+19 | **20.36** |

Figure 8: Comparisons between dense and structured FFNs with 32% parameters under the same training FLOPs. Structured FFNs are trained either with more tokens or through self-guided training to match training FLOPs. The circle size represents model FLOPs.

## 5   Conclusion

In this paper, we conducted extensive experiments investigating the use of structured matrices to parameterize FFNs in Transformers, with models scaling up to 1.3B parameters on the RefinedWeb dataset. Our primary aim was not to determine which structured matrix performs best, as this can be task-dependent, but to explore their common properties, including scaling, efficiency, and optimization challenges. We found that all of them exhibit steeper scaling curves compared to dense models. Moreover, our proposed methods, such as self-guided training, can enhance the performance across all structured matrices (e.g., `LowRank` with the novel training strategy achieves a $1.35\times$ inference speed-up with only a 0.4 increase in perplexity). To conclude, we demonstrate that structured matrices can be strong candidates to replace the dense models in architecture design by scaling studies and also reveal the challenges of applying them.

**Limitations**: `BlockDense` and `BlockShuffle` are more complicated than `LowRank`. In this work, we only explored a limited range of hyperparameter settings of them. However, since these approaches are new, we believe that further performance improvements may be possible by better tuning their hyperparameters. We primarily focused on language modeling with limited vision experiments included in the appendix. Additionally, we did not explore the optimal scaling laws for structured matrices, which may further enhance performance. We also didn't investigate models in this paper that are comparable to today's practical LLMs, such as LLaMA-3. This is not only because of the limited computing resources but also because this study is to start investigating structured parameterizations of linear layers in modern LLM architecture training. We hope our findings and solutions about scaling, efficiency, and optimization will push their usage on the industry side and in future work.

## Acknowledgments and Disclosure of Funding

We are grateful to Soham De for the insightful discussions and his valuable feedback on our work. We also sincerely thank the anonymous reviewers for their meaningful reviews and suggestions to make this better. We are also grateful to the RCP and IC cluster system administrators at EPFL for their support and assistance, especially during the project deadline. We also thank `nimble.ai` for their generous gift to CLAIRE lab, which also helped us to fund some of this research.

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

# Appendix

## A  Structured matrices

### A.1  Design choice

For `BlockDense`, we also investigate the reverse order of two projections, placing the low-rank or dense matrix first, followed by the block-diagonal matrix. However, this change surprisingly yields worse performance. For instance, on the RefinedWeb dataset, perplexity increases from 29.17 to 29.65 with Transformer-s and 2.2B training tokens. In the case of `BlockShuffle`, unlike Dao et al. [13], Zhang et al. [14] does not include a second shuffle operation to restore the original order. Taking this into account, we also experimented with removing the second shuffle operation and found almost no impact on performance. For example, with Transformer-s and -m using 32% FFN parameters, `BlockShuffle` without the second shuffle achieves perplexities of 29.89 and 21.19, respectively, compared to 29.95 and 21.12 for our adopted version. Nonetheless, we maintain the design from Dao et al. [13] for consistency.

For initialization, we follow the spectral initialization for `LowRank`, as suggested by prior works [41]. For `BlockDense` and `BlockShuffle`, motivated by Eq. (4), we propose using orthonormal initialization, setting the singular values of $U_t U_t^\top$ and $V_t V_t^\top$ to 1 at the start. Experimentally, this stabilizes training dynamics and improves the perplexity performance (Table 4). For weight decay, we tried the Frobenius decay proposed by Khodak et al. [41]; however, it did not have a clear benefit in our experiments and increased training FLOPs slightly. Hence, we adopted standard weight decay for all the structured FFNs.

Table 4: Different initialization of `BlockShuffle` and `BlockDense`, where random indicates random Gaussian initialization and orthonormal indicates orthonormal initialization. Data points are measured on the 4-layer Transformer and WikiText-103 with a learning rate of 1.0e-3.

| Method | Initialization | PPL |
|---|---|---|
| Transformer (4-layers) | random | 23.24 |
| BlockShuffle (B=4) | random | 27.24 |
| BlockShuffle (B=4) | orthonormal | **25.33** |
| BlockDense (B=2, R=128) | random | 28.25 |
| BlockDense (B=2, R=128) | orthonormal | **26.63** |

Table 5: Ablation study of self-guided training on `LowRank` trained on RefinedWeb.

| Method | PPL |
|---|---|
| **-s size** | 25.97 |
| Direct decomposition | 28.56 |
| Progressive decreasing rank | 29.35 |
| Self-guided training | **28.02** |
| Self-guided training (slower) | **27.90** |
| **-m size** | 18.29 |
| Self-guided training | **19.90** |
| Self-guided training (slower) | **19.81** |

### A.2  Results on Refinedweb dataset

For the scaling points in Fig. 4, we provide detailed results in Table 6 for easier comparison. First, all efficient parameterizations approach the baseline as model size increases. For instance, `LowRank` with 32% of the parameters has a loss gap of $0.08$ to Transformer-xl, whereas the gap is $0.12$ at the scale of Transformer-s. Moreover, `LowRank` and `BlockDense`, with 63% of the parameters in the FFN, increase the loss by only 0.02 to 0.04 while reducing total training time by approximately 15% on Transformer-xl and Transformer-l. Additionally, they accelerate training by $1.35\times$ with only a 1-point increase in perplexity on Transformer-xl with 32% of the parameters.

Although the main focus of the paper is not to compare different structured matrices but to showcase their general properties—including scaling, efficiency, and optimization—we still provide comparisons in the appendix. From Table 6, by controlling the training FLOPs and model size to be the same, `LowRank` and `BlockDense` demonstrate better performance than `BlockShuffle` in our main experiments, showing a 0.8 lower perplexity on Transformer-s and a 0.4 lower perplexity on Transformer-m. We think that for FFNs in language models, `BlockShuffle` may not be the optimal choice. However, we further compare these approaches on CIFAR-10 dataset, showing that block-diagonal matrices can serve as a good inductive bias in vision tasks Sec. A.3.

## A.3 Results on CIFAR-10 dataset

Although in our main experiments, the `BlockShuffle` performs the worst with two block-diagonal matrices, we provide experiments on the CIFAR10 dataset here, showing that when locality is highly preferred, block-diagonal matrices may perform better than low-rank matrices.

In Table 7, experiments are conducted on 5-layer MLP (MultiLayer Perceptron) and ViT models. The 5-layer MLP consists of a linear layer, batch normalization [55], and the ReLU activation function with a hidden dimension of 384. It is trained for 500 epochs with a learning rate of 1.0e-3 and a batch size of 128. For the ViT models, 12 layers with a hidden dimension of 384 are used, and they are trained for 750 epochs with a learning rate of 6.0e-4 and a batch size of 512.

Since the first layer in vision tasks typically prefers from the locality, especially in a 5-layer MLP where the image pixels are directly flattened into the input, we conducted experiments with and without structuring the first layer as `LowRank` and `BlockShuffle`. Both sets of controls, particularly when structuring the first linear layer, demonstrate that block-diagonal matrices can be beneficial for vision tasks. Specifically, replacing the first layer of the 5-layer MLP model with a block-diagonal matrix even yields better performance, as the block structure effectively groups neighboring pixels, compensating for the MLP's lack of locality. However, applying structured FFNs to the first layer of ViT can lead to significant accuracy degradation, reinforcing our decision not to use structured FFNs in the first layer in the main experiments.

Table 6: Performance of two sizes of different structured matrices with 63% and 32% of the original FFN module's parameters. We report model size, total FFN size, training tokens, training FLOPs, and training time. Note that the total structured FFN is not exactly 63% of the original because we don't replace the first FFN module. Also, the `BlockDense` is slightly smaller for -m and -xl models to ensure the rank is a multiple of 256 when matching parameters. Loss and perplexity are evaluated on a 0.5B token validation set.

| Architecture | | Model Size (M) | FFN Size (M) | Training | | | Loss | PPL |
|---|---|---|---|---|---|---|---|---|
| | | | | Tokens (B) | FLOPs | Time (h) | | |
| **Transformer-s** | | 110 | 56.62 | 2.2 | 1.69e+18 | 4.0 | 3.2569 | 25.97 |
| 63% | LowRank (R384) | 90.17 | 37.16 | 2.2 | 1.44e+18 | 3.8 | 3.3017 | **27.16** |
| | BlockDense (B2R512) | 90.17 | 37.16 | 2.2 | 1.44e+18 | 3.8 | 3.3034 | 27.20 |
| | BlockShuffle (B2) | 90.17 | 37.16 | 2.2 | 1.44e+18 | 4.2 | 3.3191 | 27.63 |
| 32% | LowRank (R192) | 73.95 | 20.94 | 2.2 | 1.22e+18 | 3.6 | 3.3748 | 29.22 |
| | BlockDense (B2R256) | 73.95 | 20.94 | 2.2 | 1.22e+18 | 3.5 | 3.3731 | **29.17** |
| | BlockShuffle (B4) | 73.95 | 20.94 | 2.2 | 1.22e+18 | 4.0 | 3.3994 | 29.95 |
| **Transformer-m** | | 335.08 | 201.33 | 6.7 | 1.55e+19 | 32.5 | 2.9062 | 18.29 |
| 63% | LowRank (R512) | 262.73 | 128.97 | 6.7 | 1.26e+19 | 29.6 | 2.9508 | **19.12** |
| | BlockDense (B4R768) | 255.19 | 121.44 | 6.7 | 1.23e+19 | 29.9 | 2.9581 | 19.26 |
| | BlockShuffle (B2) | 262.73 | 128.97 | 6.7 | 1.26e+19 | 33.1 | 2.9622 | 19.34 |
| 32% | LowRank (R256) | 202.43 | 68.68 | 6.7 | 1.01e+19 | 26.9 | 3.0251 | **20.60** |
| | BlockDense (B4R384) | 198.67 | 64.91 | 6.7 | 1.00e+19 | 27.1 | 3.0371 | 20.85 |
| | BlockShuffle (B4) | 202.43 | 68.68 | 6.7 | 1.01e+19 | 30.0 | 3.0501 | 21.12 |
| **Transformer-l** | | 729.11 | 452.98 | 14.6 | 7.03e+19 | 130.5 | 2.6594 | 14.29 |
| 63% | LowRank (R768) | 566.32 | 290.19 | 14.6 | 5.61e+19 | 113.6 | 2.6957 | **14.82** |
| | BlockDense (B2R1024) | 566.32 | 290.19 | 14.6 | 5.61e+19 | 114.3 | 2.7038 | 14.94 |
| | BlockShuffle (B2) | 566.32 | 290.19 | 14.6 | 5.61e+19 | 124.3 | 2.7021 | 14.91 |
| 32% | LowRank (R384) | 430.66 | 154.53 | 14.6 | 4.42e+19 | 100 | 2.7527 | **15.69** |
| | BlockDense (B2R512) | 430.66 | 154.53 | 14.6 | 4.42e+19 | 100.9 | 2.7570 | 15.75 |
| | BlockShuffle (B4) | 430.66 | 154.53 | 14.6 | 4.42e+19 | 110.3 | 2.7735 | 16.01 |
| **Transformer-xl** | | 1274.14 | 805.31 | 25.5 | 2.10e+20 | 352.2 | 2.5226 | 12.46 |
| 63% | LowRank (R1024) | 984.73 | 515.90 | 25.5 | 1.66e+20 | 302.2 | 2.5541 | **12.86** |
| | BlockDense (B4R1536) | 954.59 | 485.75 | 25.5 | 1.61e+20 | 298.7 | 2.5628 | 12.97 |
| | BlockShuffle (B2) | 984.73 | 515.90 | 25.5 | 1.66e+20 | 330.6 | 2.5633 | 12.98 |
| 32% | LowRank (R512) | 743.56 | 274.73 | 25.5 | 1.29e+20 | 260.2 | 2.6062 | **13.55** |
| | BlockDense (B4R768) | 728.49 | 259.65 | 25.5 | 1.27e+20 | 261.2 | 2.6204 | 13.74 |
| | BlockShuffle (B4) | 743.56 | 274.73 | 25.5 | 1.29e+20 | 284.9 | 2.6254 | 13.81 |

Table 7: Experiments on CIFAR10 and vision models, where the locality is highly preferred. The first layer column in the table indicates whether to apply structured matrices to the first FFN.

| Method | Structured first FFN | | Dense first FFN | |
|---|---|---|---|---|
| | Model size (M) | Accuracy | Model size (M) | Accuracy |
| 5-layer MLP (H=768) | 4.14 | 66.99 | 4.14 | 66.99 |
|   LowRank (R=192) | 1.63 | 64.04 | 3.26 | 65.42 |
|   BlockShuffle (B=4) | 1.63 | **67.08** | 3.26 | **65.67** |
| ViT (H=384) | 21.34 | 92.49 | 21.34 | 92.49 |
|   LowRank (R=24) | 8.29 | 89.56 | 9.38 | 92.09 |
|   BlockShuffle (B=16) | 8.29 | **90.42** | 9.38 | **92.49** |

## A.4 Results on downstream tasks

Table 8: Performance on downstream tasks under the zero-shot setting. We report the perplexity performance of the validation set of RefinedWeb. For all downstream tasks except LAMBADA, we report accuracy results. For LAMBADA, we present the results in an accuracy/perplexity format. Implementation details are put in Appendix C.

| Model | Model Size (M) | RefinedWeb | ARC (challenge) | ARC (easy) | HellaSwag | LAMBADA | PIQA |
|---|---|---|---|---|---|---|---|
| **-s size** | | | | | | | |
| Dense | 110.0 | 16.02 | 18.69 | 44.65 | 31.79 | 36.35/28.86 | 65.83 |
| Wide and Structured | 81.1 | 17.30 | 18.17 | 43.35 | 30.83 | 34.23/35.37 | 64.85 |
| **-m size** | | | | | | | |
| Dense | 353.1 | 12.34 | 22.10 | 50.80 | 38.60 | 46.65/13.92 | 70.73 |
| Wide and Structured | 219.4 | 13.38 | 20.90 | 49.12 | 36.32 | 42.46/17.60 | 69.75 |
| **-l size** | | | | | | | |
| Dense | 729.1 | 10.76 | 25.09 | 58.71 | 43.33 | 52.30/9.92 | 73.61 |
| Wide and Structured | 464.4 | 11.61 | 23.55 | 54.76 | 40.53 | 48.79/11.67 | 72.14 |

Supplementary to Fig. 5, Table 8 presents the detailed results of wide and structured networks compared to dense models on downstream tasks. These models were trained on 300B tokens and implementation details can be found in Appendix C.

# B Self-guided training

## B.1 Visualization of training dynamics

We provide additional training dynamics curves, including BlockDense and BlockShuffle, in Fig. 10 and Fig. 11, illustrating that these structures are more challenging to train compared to standard linear layers. Specifically, they exhibit greater sensitivity to learning rates and are more prone to loss spikes. To mitigate this, we apply self-guided training in the most challenging cases, which results in improved training dynamics, such as faster convergence without loss spikes.

Furthermore, we report the loss spikes observed in the large Transformer-xl model trained on the RefinedWeb dataset in Fig. 12.

## B.2 Explanation of the poor training dynamics

We observed that loss spikes occur along with large gradient norms. This motivates us to analyze the gradient updates during the backward pass of the linear $UV$. Considering $g$ as the gradient of output and $x$ as the input, the standard linear layer $W$ gradient update is $gx^\top$. For the structured parametrization, the gradients of $U$ and $V$ are $gx^\top V^\top$ and $U^\top gx^\top$, respectively. Then, for $W'$ as being the updated parameters, we will have the updates for $W$ to be:

$$\Delta W = W' - W = -lr \cdot gx^\top.$$  (3)

And the one for $UV$:

$$\Delta(UV) = -lr \cdot (UU^\top gx^\top + gx^\top V^\top V) + O(lr^2)$$  (4)

Thus, it can be seen from Eq. (4) that the projections $\boldsymbol{U}\boldsymbol{U}^\top$ and $\boldsymbol{V}^\top\boldsymbol{V}$ can disrupt the gradient $\boldsymbol{g}\boldsymbol{x}^\top$. If the norms of $\boldsymbol{U}$ and $\boldsymbol{V}$ are small, the new update vanishes faster than the original update, and in reverse, if their norm is large, the update blows up, leading to unstable training.

To be specific, we calculate the spectral norm of $\boldsymbol{U}\boldsymbol{U}^\top$ and $\boldsymbol{V}^\top\boldsymbol{V}$ and use this to indicate the maximum scale the matrix can stretch a vector. Fig. 13 shows that the largest singular value can vary significantly, being either much greater than or less than 1, depending on the shape of the weight (input dimension, rank, number of blocks) and magnitude. Interestingly, very structured FFNs with small ranks or many blocks tend to have smaller spectral norms, while others have larger ones. This corresponds to the phenomenon in Fig. 10 and Fig. 11, where smaller FFN are prone to slower convergence and larger ones to loss spikes.

An alternative intuitive perspective of this issue is to realize that learning with structured matrices has additional redundant degrees of freedom brought by symmetry. For example, to increase the norm of a feature, it can increase either the corresponding weights $\boldsymbol{U}$ or $\boldsymbol{V}$. Given that gradient descent makes this decision independently, it will overshoot and make learning less well-behaved.

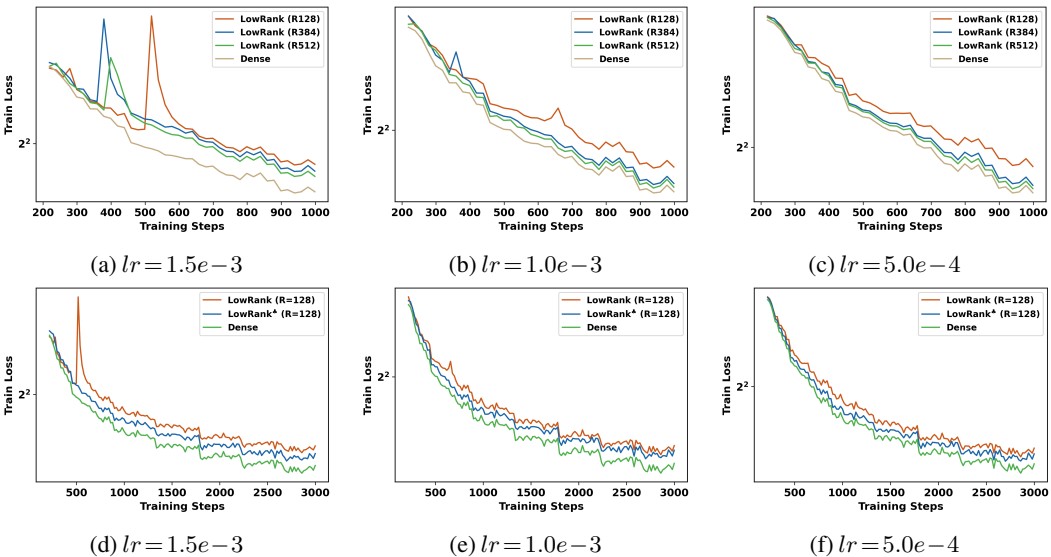

Figure 9: (a-c): Training dynamics of `LowRank` with different ranks and the dense model under different hyper-parameters. Data points are measured on a 4-layer Transformer with model width 768 and WikiText-103. We zoom into the beginning of training for clearer observations. (d-f): Training dynamics of `LowRank` and the self-guided training. The self-guided training overcomes the loss spikes and makes the training faster. We show the whole training curve to indicate its success. $R$ indicates the rank of low-rank matrices.

### B.3 Design choice

Experiments in this part are conducted on Transformer-s and Transformer-m with ranks 192 and 256 in Table 5, respectively. We apply self-guided training during the first half of the training process.

First, we compare stochastic self-guided training with the static version. The stochastic and faster version in both model sizes brings about a 0.1 perplexity increase while reducing computation by half. Second, other techniques are compared. Dense layer decomposition, which decomposes the weight directly at the midpoint of training, is examined. This approach can lead to abrupt loss increases in training curves, resulting in worse performance. Strategies incrementally reducing rank require a feasible and complex change strategy and fail to address the inconsistent gradient update problem, thus still suffering from poor results Table 5.

Generally speaking, our method stands out due to its flexibility, simplicity, and efficiency. Eq. (1) makes it adaptable to any efficient linear parametrization without special constraints, while progressive rank reduction and direct decomposition require a feasible solution to evolve. Our guided initialization allows its usage in various stages of training without the need for a well-trained

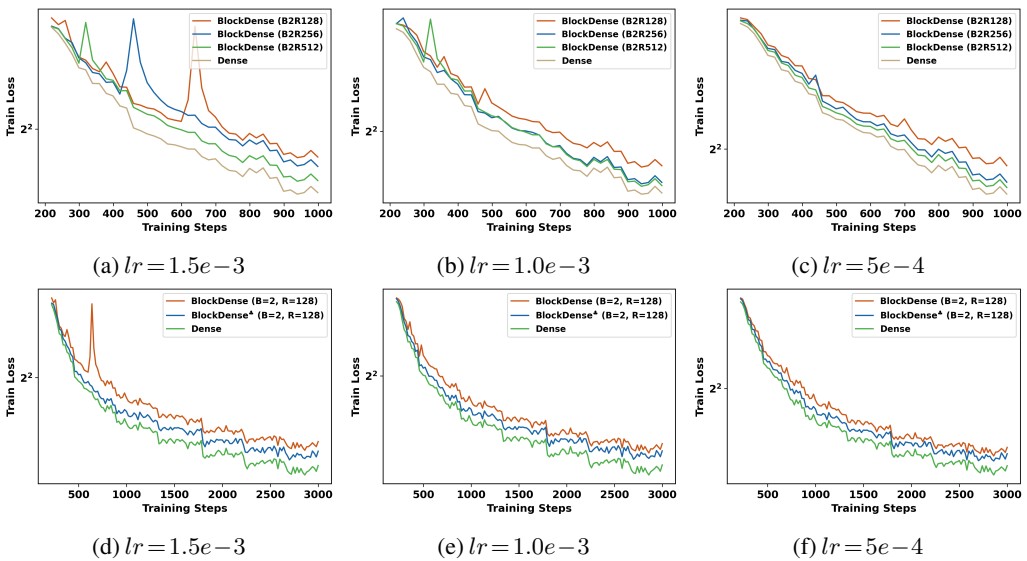

Figure 10: (a-c): Training dynamics of `BlockDense` with 2 blocks and different ranks and the dense model under different hyper-parameters. (d-f): For `BlockDense` ($B=2, R=128$), training dynamics of self-guided training indicated by ♣. Other settings follow Fig. 9.

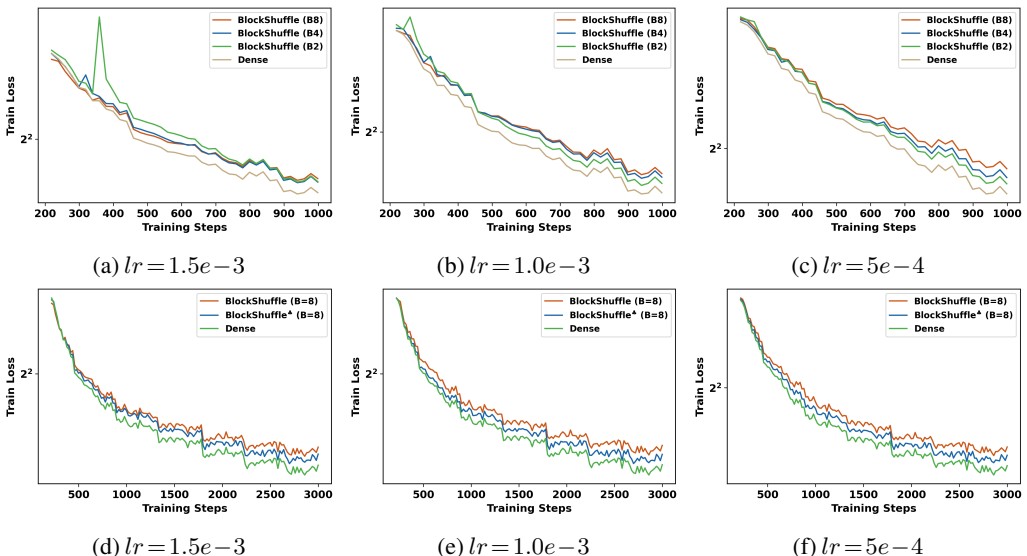

Figure 11: (a-c): Training dynamics of `BlockShuffle` with different numbers of blocks and the dense model under various hyper-parameters. (d-f): For `BlockShuffle` ($B=8$), loss curves of self-guided training indicated by ♣. Other settings follow Fig. 9. Note that the training dynamics of different structured matrices are not comparable here because their sizes are not controlled to be the same.

teacher. It is simple because Eq. (1) provides only one smooth transition. It is efficient due to the layer-specific definition in Eq. (1) and stochastic computation in Eq. (2).

## B.4   Results on Refinedweb Dataset

Table 9 provides more comprehensive results of self-guided training, supplementing Sec. 4.4. For Transformer-s and -m, we first present the results of applying self-guided training to the first half of training in the second row of each table block, demonstrating its effectiveness across all three structured matrices. Additionally, by controlling for equal training FLOPs as described in Sec. 4.4,

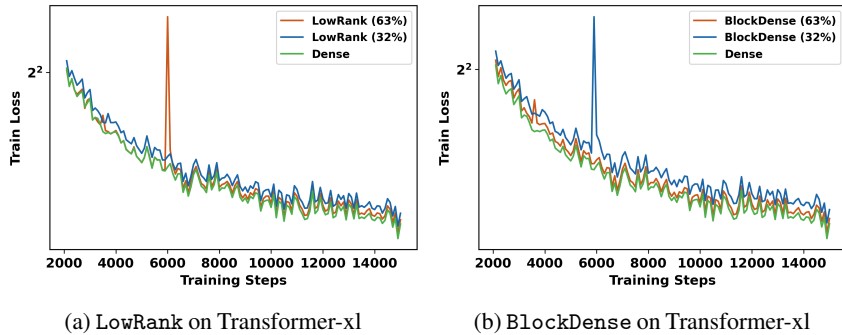

(a) `LowRank` on Transformer-xl      (b) `BlockDense` on Transformer-xl

Figure 12: Loss curves of Transformer-xl. Structured FFN with 32% parameters exhibits slower convergence. Also, there exist loss spikes.

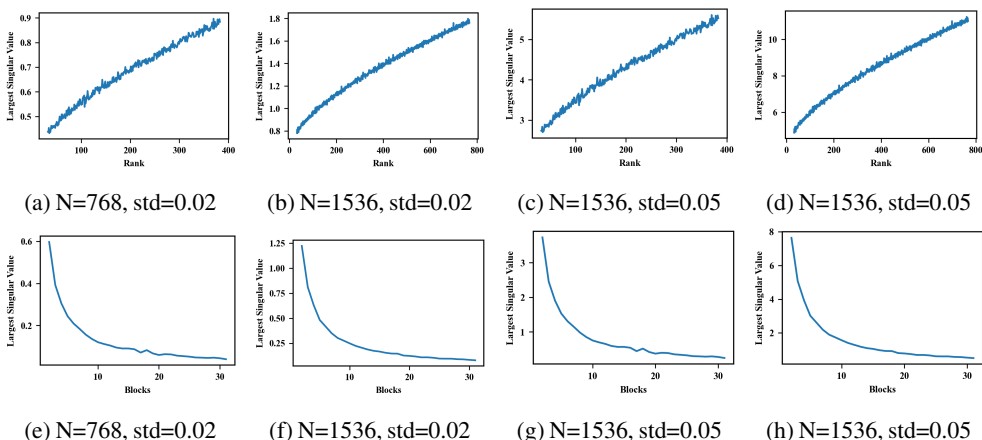

(a) N=768, std=0.02    (b) N=1536, std=0.02    (c) N=1536, std=0.05    (d) N=1536, std=0.05

(e) N=768, std=0.02    (f) N=1536, std=0.02    (g) N=1536, std=0.05    (h) N=1536, std=0.05

Figure 13: Spectral norm of the matrix $\boldsymbol{V}^{\top}\boldsymbol{V}$, where in (a-c), $\boldsymbol{V}$ is the low-rank matrix and in (d-f), $\boldsymbol{V}$ is the block-diagonal matrix. N presents the input dimension of the weight. Std indicates the standard deviation value of the normal distribution from which we sample weight elements.

we show that with self-guided training, all three structured FFNs with 32% parameters incur only a 0.4-0.6 increase in perplexity compared to the baseline, while benefiting from a smaller memory footprint and faster speed at inference time. To further illustrate this, we plot these points with the same training FLOPs in Fig. 14, and specifically for `LowRank` in Fig. 8, highlighting that self-guided training achieves comparable performance to training on more tokens.

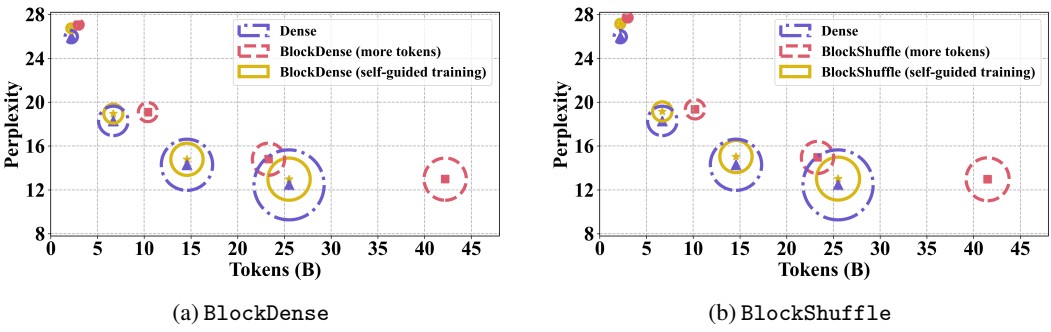

(a) `BlockDense`      (b) `BlockShuffle`

Figure 14: Performance between dense Transformer, Structured FFN (`BlockDense` and `BlockShuffle`, `LowRank` in Fig. 8) with 32% parameters either trained with self-guided training or more tokens across four sizes. Circle size indicates model FLOPs. To enable straightforward comparison, we controlled their training FLOPs to be the same.

Table 9: Performance of self-guided training indicated by ♣ on Structured FFN with 32% parameters under the same training FLOPs. We also include the structured FFN trained on more different tokens as a highly advanced baseline. Model FLOPs are calculated on one sample with 1024 sequence length.

| Architecture | Model Size (M) | FFN Size (M) | Model FLOPs (G) | Training | | Loss | PPL |
|---|---|---|---|---|---|---|---|
| | | | | Tokens (B) | FLOPs | | |
| **Transformer-s** | 110.0 | 56.6 | 262.9 | 2.2 | 1.69e+18 | 3.2569 | 25.97 |
| LowRank | | | | 2.2 | 1.22e+18 | 3.3748 | 29.22 |
| LowRank♣ | 74.0 | 20.9 | 189.8 | 2.2 | 1.39e+18 | 3.3329 | **28.02** |
| LowRank | | | | 3.0 | 1.69e+18 | 3.2928 | 26.92 |
| LowRank♣ | | | | 2.2 | 1.69e+18 | 3.2866 | **26.75** |
| BlockDense | | | | 2.2 | 1.22e+18 | 3.3731 | 29.17 |
| BlockDense♣ | 74.0 | 20.9 | 189.8 | 2.2 | 1.39e+18 | 3.3338 | **28.04** |
| BlockDense | | | | 3.0 | 1.69e+18 | 3.2982 | 27.06 |
| BlockDense♣ | | | | 2.2 | 1.69e+18 | 3.2856 | **26.73** |
| BlockShuffle | | | | 2.2 | 1.22e+18 | 3.3994 | 29.95 |
| BlockShuffle♣ | 74.0 | 20.9 | 189.8 | 2.2 | 1.39e+18 | 3.3583 | **28.74** |
| BlockShuffle | | | | 3.0 | 1.69e+18 | 3.3218 | 27.71 |
| BlockShuffle♣ | | | | 2.2 | 1.69e+18 | 3.3011 | **27.14** |
| **Transformer-m** | 335.1 | 201.3 | 788.7 | 6.7 | 1.55e+19 | 2.9062 | 18.29 |
| LowRank | | | | 6.7 | 1.01e+19 | 3.0251 | 20.60 |
| LowRank♣ | 202.4 | 68.7 | 517.0 | 6.7 | 1.21e+19 | 2.9907 | **19.90** |
| LowRank | | | | 10.2 | 1.54e+19 | 2.9359 | 18.84 |
| LowRank♣ | | | | 6.7 | 1.55e+19 | 2.9310 | **18.75** |
| BlockDense | | | | 6.7 | 1.00e+19 | 3.0371 | 20.85 |
| BlockDense♣ | 198.7 | 64.9 | 509.3 | 6.7 | 1.19e+19 | 3.0008 | **20.10** |
| BlockDense | | | | 10.4 | 1.55e+19 | 2.9491 | 19.09 |
| BlockDense ♣ | | | | 6.7 | 1.55e+19 | 2.9420 | **18.95** |
| BlockShuffle | | | | 6.7 | 1.01e+19 | 3.0501 | 21.12 |
| BlockShuffle♣ | 202.4 | 68.7 | 517.0 | 6.7 | 1.21e+19 | 3.0135 | **20.36** |
| BlockShuffle | | | | 10.2 | 1.54e+19 | 2.9627 | 19.35 |
| BlockShuffle♣ | | | | 6.7 | 1.55e+19 | 2.9525 | **19.15** |
| **Transformer-l** | 729.1 | 453.0 | 1646.9 | 14.6 | 7.03e+19 | 2.6594 | 14.29 |
| LowRank | | | | 14.6 | 4.42e+19 | 2.7527 | 15.69 |
| LowRank | 430.7 | 154.5 | 1035.6 | 23.3 | 7.03e+19 | 2.6917 | 14.76 |
| LowRank♣ | | | | 14.6 | 7.01e+19 | 2.6850 | **14.66** |
| BlockDense | | | | 14.6 | 4.42e+19 | 2.7570 | 15.75 |
| BlockDense | 430.7 | 154.5 | 1035.6 | 23.3 | 7.03e+19 | 2.6946 | 14.80 |
| BlockDense♣ | | | | 14.6 | 7.01e+19 | 2.6941 | **14.79** |
| BlockShuffle | | | | 14.6 | 4.42e+19 | 2.7735 | 16.01 |
| BlockShuffle | 430.7 | 154.5 | 1035.6 | 23.3 | 7.03e+19 | 2.7053 | 14.96 |
| BlockShuffle♣ | | | | 14.6 | 7.01e+19 | 2.7104 | **15.04** |
| **Transformer-xl** | 1274.1 | 805.3 | 2814.3 | 25.5 | 2.10e+20 | 2.5226 | 12.46 |
| LowRank | | | | 25.5 | 1.29e+20 | 2.6062 | 13.55 |
| LowRank | 743.6 | 274.7 | 1727.7 | 41.5 | 2.10e+20 | 2.5464 | 12.76 |
| LowRank♣ | | | | 25.5 | 2.10e+20 | 2.5539 | **12.86** |
| BlockDense | | | | 25.5 | 1.27e+20 | 2.6204 | 13.74 |
| BlockDense | 728.5 | 259.7 | 1696.8 | 42.2 | 2.10e+20 | 2.5590 | 12.92 |
| BlockDense♣ | | | | 25.5 | 2.10e+20 | 2.5637 | **12.98** |
| BlockShuffle | | | | 25.5 | 1.29e+20 | 2.6254 | 13.81 |
| BlockShuffle | 743.6 | 274.7 | 1727.7 | 41.5 | 2.10e+20 | 2.5623 | 12.97 |
| BlockShuffle♣ | | | | 25.5 | 2.10e+20 | 2.5678 | **13.03** |

# C  Implementation details

**Model**  Architecture details are provided in Table 10. We consider four baseline Transformer sizes, ranging from 110M to 1.3B parameters, with widths from 768 to 2048. For structured models, we first adopt two configurations as described in the main paper, reducing the FFN module to 63% and 32% of its original parameters by adjusting the rank $R$ and number of blocks $B$. Only $R$ and $B$ values specifically associated with structured matrices are modified, as seen in Table 6. Then, for more comparable results, we consider wide and structured networks, where the attention module is also structured by reducing the attention heads. We also present the transformer with GQA version [2] here, configured with 256 dimensions for the KVCache [9] and an enlarged FFN intermediate dimension following [8]. Based on this GQA version, we apply `LowRank` matrices to the FFN module with a rank half of the model or FFN width and use a smaller attention inner dimension to further reduce the parameters of the attention module. This allows us to maintain the parameter ratio between the attention and FFN modules.

Note that in all experiments, we do not apply structured matrices to the first FFN module, as doing so can lead to non-negligible performance loss in models on shallow networks. For dense models, we use Gaussian random weight initialization with a standard deviation of 0.02. For structured matrices, spectral initialization is applied for `LowRank`, and orthonormal initialization for the other two, based on initial experiments.

Table 10: Detailed configurations of the baseline Transformers, along with those using GQA [2] and wide and structured networks. The latter two are employed in the scaling study in Sec. 4.2. For other structured models which have 63% and 32% of the original FFN parameters, we adjust only the rank and number of blocks for each method and put the configuration directly in Table 6. **Width** denotes the model width or the input and output dimensions of the attention and FFN modules. **Intermediate dim.** refers to the intermediate dimension of the FFN. **Attention dim.** specifies the dimension used in scaled-dot product attention. **KV dim.** represents the dimension used for KVCache, as selected according to Team et al. [9].

| Model | Size (M) | Layers | Width | Intermediate dim. | Attention dim. | KV dim. |
|---|---|---|---|---|---|---|
| Transformer-s | 110.0 | 12 | 768 | 3072 | 768 | 768 |
| Transformer-s (GQA) | 110.0 | 12 | 768 | 3584 | 768 | 256 |
| Wide and Structured (R=384) | 81.1 | 12 | 768 | 3584 | 512 | 256 |
| Transformer-m | 335.1 | 24 | 1024 | 4096 | 1024 | 1024 |
| Transformer-m (GQA) | 335.1 | 24 | 1024 | 4864 | 1024 | 256 |
| Wide and Structured (R=512) | 219.4 | 24 | 1024 | 4864 | 512 | 256 |
| Transformer-l | 729.1 | 24 | 1536 | 6144 | 1536 | 1536 |
| Transformer-l (GQA) | 729.1 | 24 | 1536 | 7424 | 1536 | 256 |
| Wide and Structured (R=768) | 464.4 | 24 | 1536 | 7424 | 768 | 256 |
| Transformer-xl | 1274.1 | 24 | 2048 | 8192 | 2048 | 2048 |
| Transformer-xl (GQA) | 1274.1 | 24 | 2048 | 9984 | 2048 | 256 |
| Wide and Structured (R=1024) | 799.6 | 24 | 2048 | 9984 | 1024 | 256 |

**Dataset**  We use the RefinedWeb dataset [48], a carefully curated subset of CommonCrawl, optimized for filtering and deduplication, providing 600B tokens for public use. Due to its large size, we shuffle, extract, and tokenize it in advance. To manage memory efficiently, token IDs are stored using `np.memmap`, preventing the need to load all data into CPU memory simultaneously. The maximum sequence length is set to 1024. Following scaling laws [1], we allocate tokens at 20 times the number of parameters for each baseline model in all experiments, except for the 300B token training.

Table 11: **Basic training configuration** used in all experiments except for the overtraining regime with 300B tokens. Note that we apply the same global batch size (**Batch**) and the same peak learning rates (**LR**) to both dense and structured models to avoid hyperparameter search. The hyperparameter values are selected based on Zhang et al. [50], Gu and Dao [49].

| Model | Tokens | Batch | LR |
|---|---|---|---|
| **-s size** | | | |
| Dense | 2.2B | 512 | 6.0e-4 |
| Structured | | | |
| **-m size** | | | |
| Dense | 6.7B | 512 | 3.0e-4 |
| Structured | | | |
| **-l size** | | | |
| Dense | 14.6B | 512 | 2.5e-4 |
| Structured | | | |
| **-xl size** | | | |
| Dense | 25.5B | 512 | 2.0e-4 |
| Structured | | | |

Table 12: **Training configuration for 300B token training.** Different studies [49, 52, 47, 51] employ very different learning rates in this setting, which also differ from training-compute scaling studies [1]. To avoid extensive tuning, we follow the hyperparameter scaling rule of Transformer proposed by Bi et al. [53], determining batch size and learning rate based on training FLOPs.

| Model | Size (M) | Batch | LR |
|---|---|---|---|
| **-s size** | | | |
| Dense | 110.0 | 1280 | 9.1e-4 |
| Structured | 81.1 | 1024 | 9.6e-4 |
| **-m size** | | | |
| Dense | 335.1 | 1792 | 7.8e-4 |
| Structured | 219.4 | 1536 | 8.4e-4 |
| **-l size** | | | |
| Dense | 729.1 | 2304 | 7.1e-4 |
| Structured | 464.4 | 2048 | 7.6e-4 |
| **-xl size** | | | |
| Dense | 1274.1 | 2816 | 6.6e-4 |
| Structured | 799.6 | 2304 | 7.1e-4 |

**Training**  We use A100 80G GPUs for training and evaluation, employing mixed precision (`bfloat16` and `float32`) with `torch.cuda.amp` to accelerate training. Training FLOPs are calculated following Megatron [54], including all matrix multiplications.

Different hyperparameters are used for 300B token training and other experiments. For basic training in training FLOPs scaling and self-guided training studies, configurations are listed in Table 11. Hyperparameter values are selected based on the scaling law studies of Zhang et al. [50], Gu and Dao [49], where we use the same learning rates and global batch size for both dense and structured models. Additional details include the AdamW optimizer with 0.1 weight decay, betas of [0.9, 0.999], cosine annealing learning rate scheduler with 10% linear warm-up, and $0.1\times$ minimum value. Dropout is set to 0.0, and gradient clipping to 1.0.

In the overtraining regime where the training duration is super long, however, smaller betas [0.9, 0.98] are required for stable training, even for baseline Transformers. Previous studies [49, 52, 47, 51] adopt very different learning rates in this setting, differing from training-compute scaling studies [1]. To avoid extensive searching, we follow the hyperparameter scaling rule of Transformer proposed by Bi et al. [53], determining the global batch size and learning rate based on training FLOPs. Specifically, batch size is defined by $0.3118 \times (\texttt{training FLOPs}^{-0.125})$, and learning rate by $0.2920 \times (\texttt{training FLOPs}^{0.3271})$, giving the results in Table 12. It can be seen that our wide and structured models trained on 300B tokens will use a slightly higher learning rate and smaller batch size compared to the larger Transformer.

**Efficiency**  To enhance training and inference efficiency, our code is based on `PyTorch` but incorporates optimized CUDA kernels. We leverage Flash Attention [25], fast LayerNorm, and rotary embeddings from TransformerEngine [56], along with fused operations including bias and GeLU. For inference speed testing, we use `bfloat16`. These techniques are consistently applied to all models to ensure fair latency and throughput comparisons.

## D  Broader Impacts

Enhancing the efficiency of Large Language Models (LLMs) can significantly reduce computational resources and energy consumption, benefiting the environment and democratizing access to advanced AI technologies. However, increased efficiency could also lead to greater dissemination of disinformation and the creation of deepfakes, posing risks to public trust and security and potentially reinforcing existing biases that impact specific groups unfairly. This research aims to promote the responsible development and deployment of LLMs, maximizing societal benefits while acknowledging potential harms.

