# OpenReview forum: "Building on Efficient Foundations: Effective Training of LLMs with Structured Feedforward Layers"
_NeurIPS.cc/2024/Conference — NeurIPS 2024 poster_

### Official Review · Reviewer_SNmM · 2024-07-10

**Soundness:** 3
**Presentation:** 3
**Contribution:** 3
**Rating:** 5
**Confidence:** 4

**Summary:**

In order to improve the efficiency of Large Language Models, the authors explore the use of three structured approximations in the FFN blocks of the Transformer: LowRank, BlockShuffle, and BlockDense. They consider both pre-training and decoding, which have distinct requirements and bottlenecks and a range of sizes from 110M to 1.3B. Furthermore, they introduce self-guided training, which uses a dense matrix as a residual component, which is then annealed. The proposed method achieves a 2.5x speed-up at a 0.4 PPL increase under the same training FLOPs.

**Strengths:**

- The research topic of improving the efficiency of LLMs and, therefore, making them more affordable is crucial and timely. The authors chose to study the FFN bottleneck, which gets worse as models scale, making the research even more relevant as LLMs become larger.
- The authors conducted extensive experiments and ablations, notably on the model size, pre-training tokens, FFN width, batch size, and learning rate. Additionally, they provided scaling laws.
- The experimental setup is common and modern. Notably, they used RefineWeb, RoPE, GeLU, FlashAttention, GQA, and tuned betas.
- The paper is well-written and easy to follow.

**Weaknesses:**

- The models were trained between 2B and 25B tokens, amounting to a few thousand steps, which is not enough for a language model to converge even at a smaller scale. Therefore, one cannot conclude whether the proposed method is competitive or degrades the performance compared to the baseline at (or close to) convergence.
- There is no downstream task evaluation of the models. Only the validation perplexity is reported.
- The simple baseline where the FFN is not expanded (`d_model -> d_model -> d_model`) is missing.

**Questions:**

- The scaling law seems to indicate that the proposed approach leads to worse performance per FLOPs. If so, what is the benefit of the proposed approach?
- Can you conduct one experiment at 300B tokens to show that your methods remain competitive in terms of performance at (or close to) convergence? e.g., the best-performing structured approximation with self-guided training against a vanilla vaseline.
- Can you add the baseline where the FFN is not expanded as well as some evaluation on downstream tasks?

**Limitations:**

I would like the authors to acknowledge the limitations of the model size and number of tokens in comparison to state-of-the-art language models (such as LLAMA3), as their results may not be applicable at scale.

---

> ### Author Rebuttal · Authors · 2024-08-07
>
> We would like to thank the reviewer for the thoughtful review and insights on this paper. We have added the necessary experiments as suggested and provided a detailed response below. We hope our response addresses the reviewer’s questions.
>
> * **Q1**: The models were trained between 2B and 25B tokens, which is not enough for a language model to converge even at a smaller scale. Therefore, one cannot conclude whether the proposed method is competitive or degrades the performance compared to the baseline at (or close to) convergence. Can you conduct one experiment at 300B tokens ...?
>
>    **A1** : Thanks for your good suggestion. On the one hand, we follow the Chinchilla scaling law paper to allocate training tokens. This allows us to compare the scaling curves of these parameterizations with the optimal curve of the dense model and indicate their effectiveness at larger scales. On the other hand, to directly demonstrate that they maintain performance at the overtraining regime, we trained Transformers (110M) on 100B tokens within this short rebuttal period.
>
>   | Method| FFN size (M) | 2.2B Tokens (optimal dense scaling law) | 100B Tokens (overtraining regime)|
>   |-|-|-|-|
>   | | |Loss/ Perplexity | Loss/Perplexity|
>   |Transformer-s| 57|3.2569/25.67| 2.8143/16.68|
>   |LowRank | 21|3.3748/29.22|2.9256/18.65|
>   |BlockDense| 21| 3.3731/29.17|2.9239/18.61|
>   |BlockShuffle | 21| 3.3994/29.95|2.9413/18.94|
>
>  By comparing the performance between models trained with 2.2B and 100B tokens, it can be seen that these structured parameterizations maintain or slightly reduce the loss gap on 100B tokens, indicating that they’re still competitive at convergence.
>
> * **Q2** : I would like the authors to acknowledge the limitations of the model size and number of tokens in comparison to state-of-the-art language models (such as LLAMA3), as their results may not be applicable at scale.
>
>   **A2** : Thanks for pointing this out. We will definitely add the limitation in the revision that we didn’t investigate models in this paper that are comparable to today’s practical LLMs, such as LLaMA-3. This is not only because of the limited computing resources but also because this study is to start investigating structured parameterizations of linear layers in modern LLM architecture training. We hope our findings and solutions about scaling, efficiency, and optimization will push their usage on the industry side and in future work.
>
> * **Q3**: There is no downstream task evaluation of the models.
>
>   **A3**: We train the Transformer-s models on 100B tokens and evaluate their zero-shot downstream performance. Table 1 in the response PDF shows that the results are consistent with the validation perplexity of pre-training. Structured parameterizations with 32% FFN parameters incur only about 0.9-1.4 accuracy loss on the downstream evaluations.  Besides, we also provided similar good results of Transformer-xl trained on 26B tokens in Table 2 of the response PDF and the self-guided training consistently improves downstream performance by reducing training loss.
>
> * **Q4**: The scaling law seems to indicate that the proposed approach leads to worse performance per FLOPs. If so, what is the benefit of the proposed approach?
>
>   **A4**: Compared to the dense model (optimal trade-off), structured matrices have the advantage of utilizing training FLOPs more effectively, potentially reaching lower loss with fewer parameters (see Figure 1 and Table 2 in the original paper). We will clarify this point more clearly in the revision. Specifically,
>     * In Figure 1 of the general response PDF, we apply a linear fit to the scaling points for better illustration. We train the dense and structured models on the same amount of tokens. By fixing the training FLOPs, structured matrices have fewer parameters and eventually achieve very close or even slightly better (e.g., LowRank) loss in Figure 1. Given their steeper scaling curves, we can also expect noticeably lower loss and fewer parameters for structured parameterizations per FLOP when the x-axis is further extended.
>     * In the "Wide and Structured Network" section of the paper, we also apply existing efficient techniques to the attention module to further optimize the use of training FLOPs. In most experiments, we only structure the FFN module to simplify the study, which negatively increases the attention module's impact on the overall architecture. By making the whole network wide and structured, we demonstrate that even with a medium-sized Transformer (335M), we achieved 0.5 lower perplexity with only 252M parameters under the same training FLOPs.
>
>   Moreover, the scaling curves of the structured matrices can be further optimized by finding a better trade-off. The good scaling behavior makes them strong candidates for future architecture design.
>
> * **Q5**: The simple baseline where the FFN is not expanded (d_model -> d_model -> d_model) is missing.
>
>   **A5** : We understand that this baseline is to reduce the intermediate size of the FFN, resulting in a smaller FFN as well. However, we chose not to pursue this approach because our investigation focuses on structured linear transformations (e.g., low-rank or block-diagonal matrices), as stated in the introduction.
>      * First, these structured matrices serve as plug-and-play replacements for the dense layer. This allows us to extend our findings to dense layers in other architectures easily. Also, maintaining the input and output dimensions doesn't affect the design of other components.
>     * Second, techniques like pruning cannot function as plug-and-play transformations, as they may require complex dimension changes and impact other components in the network. Moreover, they fall outside the scope of our current context and are left for future work.
>
>    In summary, we excluded FFN changes to simplify our explorations. We believe these changes are also orthogonal and can be combined with the structured linear transformations.

---

> > ### Comment · Reviewer_SNmM · 2024-08-09
> >
> > Thank you for the detailed rebuttal. Overall, I am pleased with the answers and clarifications provided, and I’ve adjusted my score accordingly. I am not convinced by the arguments provided against the baseline where the FFN is not expanded, that is, the FFN remains dense with both layers set to the same size (d_model = dim_feedforward). I strongly recommend that the authors include this baseline.
> >
> > **A1.** I am satisfied with the additional experiments at 100B tokens, given the computational cost of pre-training.
> >
> > **A2.** I appreciate the commitment.
> >
> > **A3.** I am pleased with the downstream evaluations, considering the limited time for the rebuttal, although I still believe additional experiments are necessary to make a strong case.
> >
> > **A4.** Thank you for the provided figures on the scaling law. It is now clear to me that structured FFNs could outperform dense FFNs at larger scales.
> >
> > **A5.** I disagree with the response on this point. I believe it is important to include a baseline where the FFN remains dense, as in the vanilla Transformer, but the second layer is the same size as the first. This baseline requires only setting `torch.nn.Transformer(d_model=512, dim_feedforward=512, ...)` in PyTorch. This baseline requires no implementation since only a single parameter needs to be changed, and the FFN remains structured (dense).

---

> > > ### Author Response · Authors · 2024-08-13
> > > **Response to Reviewer SNmM**
> > >
> > > Thanks for your quick reply and raised score. We ran the experiments that the reviewer asked and added the validation loss/perplexity results below. Baseline 1 indicates the standard Transformer with (d_model -> 4 * d_model -> d_model) FFN dimensions. Baseline 2 indicates (d_model -> d_model -> d_model) for FFN module.
> > >
> > > || Transformer-s  | Transformer-m | Transformer-l | Transformer-xl | Loss gap between -xl and -s  | Slope of the scaling curve |
> > > |-|-|-|-|-|-|-|
> > > | Baseline1 (100% FFN Params.) | 3.2569/ 25.97  | 2.9062/18.29  | 2.6594/14.29  | 2.5226/12.46   | 0.7343 | -0.3549  |
> > > | Baseline 2 (25% FFN Params)  | 3.3695/ **29.06**  | 3.0402/20.91  | 2.7862/16.22  | 2.6470/ 14.11  | 0.7225 | -0.3636  |
> > > | LowRank (32% FFN Params.     | 3.3748/29.22   | 3.0251/ **20.60**  | 2.7527/ **15.69**  | 2.6062/ **13.55**  | 0.7686 | -0.3852|
> > >
> > > * Firstly, LowRank performs much better at larger scales compared to Baseline 2, with a bigger loss gap between -xl and -s models and a steeper scaling curve. We think this is because reducing the intermediate dimension directly reduces the parameters, while the structured parameterization of the dense layer, like LowRank, is an approximation of the dense transformation. Thus, LowRank seems to perform better as the scale of the model increases.
> > >
> > > * Secondly, we think that the reason Baseline 2 works slightly better than LowRank in the -s size is that, at small model scales, the loss caused by its optimization challenges cannot be completely mitigated by the benefits of better structural design. By alleviating the optimization challenges (e.g., applying self-guided training in the first half of the total training), performance improves from 29.22 to 28.02.
> > >
> > > Finally, structured parameterization is a flexible technique that can replace any dense linear layer directly without altering its shape and can also be combined with other methods like reducing the intermediate state size as Baseline 2.

---

### Official Review · Reviewer_5PEQ · 2024-07-11

**Soundness:** 3
**Presentation:** 3
**Contribution:** 3
**Rating:** 5
**Confidence:** 2

**Summary:**

This paper mainly focues on using structured matrices to substitute dense matrices in FFNs for training from scratch tasks. The authors propose BlockDense as a combination of low-ranked dense and block diagonal matrices (Figure 2), and to address the loss spike issues in low-ranked training process, the authors propose the "self-guided training" where we combine the efficient training with parameterized linear dense layer as $o = \alpha W x + (1 - \alpha) U (V x)$ to regularize / alleviate the optimization difficulties from saddle points and pathologies.

The authors conduct experiments on transformers with sizes from 110M to 1.3B and RefinedWeb dataset. The authors focus on the latency vs. batch size, width, etc. for online decoding and validation loss vs. FLOPs. The baselines are low-ranked matrices, block shuffle (Monarch decomposition) matrices, and dense matrices for FFNs training, and the authors show Block Dense in general achieve lower PPL than Block Shuffle on the same training budget (training FLOPs).

**Strengths:**

This paper use structured matrices as a hardware-friendly efficient training method. Comprehensive latency benchmarking results are quite informative for deploying the BlockDense (or other methods) in practice.

This paper also provide comprehensive training-from-scratch experiments with medium-sized transformers.

The writing is clear to follow, and Figure 2 is quite helpful for understanding BlockDense parameterization.

**Weaknesses:**

Self-guided training is effectively adding an regularization term that combines $U$ and $V$, and we need to compare it against other regularization approaches (e.g. spectral regularization) to position the effectiveness of self-guided training with related works. Such comparison is missing from the paper (missing on both Figure 3 and Table 3) and it is hard to assess the effectiveness of self-guided training as a regularization method.

In addition, I don't find the advantage of BlockDense over low ranked matrices. It seems that low-ranked matrices also have low latency (Figure 4) and low validation loss / perplexity on the same training budget (Table 3). The optimization challenge of low-ranked matrices can also be alleviated by self-guided training according to Figure 3.

It is also hard to identify whether the scaling law of BlockDense is necessarliy better than the dense matrices in Figure 1a. The orange / green line appear to be nearly parallel to the purple line in both $10^{18}-10^{19}$ and $\sim 10^{20}$ segments. More quantitative analyses are needed to justify the statement on line 52 "Interestingly, scaling curves in Fig. 1a indicate that structured matrices yield steeper loss scaling curves than the traditional Transformer at its optimal trade-off".

I currently consider the first 2 weaknesses as major (insufficient comparison and justification) and I vote for borderline reject. I am happy to raise my scores if the above concerns are addressed during the rebuttal period.

EDIT: upon reading the rebuttal, I raise my score to the borderline accept.

**Questions:**

Could you compare self-guided training with other regularization methods?

Could you justify why BlockDense is a better efficient training method than low-ranked training?

Could you provide more quantitative analyses on the scaling laws of block dense, low-ranked matrices, block shuffle, and dense matrices (actual slope, confidence interval, etc.)?

**Limitations:**

There are no other limitations. All weaknesses have been listed above.

---

> ### Author Rebuttal · Authors · 2024-08-07
>
> We would like to thank the reviewer for providing a constructive review and detailed comments.
>
> Before responding in detail, firstly we would like to clarify our paper's focus. We investigate the performance of three structured matrices in modern LLM training from efficiency, optimization, and scaling aspects and provide concrete ablations to improve our scientific understanding of these methods. As a result, 1) we explore BlockDense to cover a broader space of possible parametrization, but not claiming that it is the best candidate, and we found that it underperforms when compared to LowRank in some experimental settings; 2) the challenges and proposed solutions (e.g., self-guided training, pre-merge, and scaling curves) are found or proposed for all the parameterizations.
>
> * **Q1**: In addition, I don't find the advantage of BlockDense over low ranked matrices. It seems that low-ranked matrices also have low latency (Figure 4) and low validation loss .... The optimization challenge of low-ranked matrices can also be alleviated by self-guided training.
>
>    **A1**: As stated earlier, this paper is not claiming that BlockDense is the best candidate but aims to investigate the performance of three structured matrices from several aspects within the FFN module of Transformers. Our findings and solutions like self-guided training also work for all of them.
>
>   We propose BlockDense for several reasons:
>     * It is a natural intermediate between LowRank and BlockShuffle, combining low-rank projection with a block-diagonal matrix.
>     * It shows similar results to LowRank in our various latency tests and has good validation loss, which helps it to cover a broader space of possible parameterizations.
>     * This parameterization might be more beneficial in other domains or architectures, as we later found that BlockShuffle works better in vision tasks.
> * **Q2**: Self-guided training is effectively adding a regularization term that combines U and V, and we need to compare it against other regularization approaches (e.g. spectral regularization) to position the effectiveness of self-guided training with related works. Such comparison is missing from the paper (missing on both Figure 3 and Table 3) and it is hard to assess the effectiveness of self-guided training as a regularization method.
>
>   **A2** : We provide the clarification about self-guided training and comparison with other regularization techniques below.
>     * We would like to first clarify that self-guided training is not a typical regularization technique. It does not explicitly constrain or normalize U and V but leverages the dense matrix W to shape the representation learned by structured matrices. Specifically, learning W is unaffected by the additional saddles and pathologies introduced by the structured parametrization, allowing it to learn faster by discovering good specialized features. Then, by decaying the residual contribution of the dense matrix, W can guide the training and transfer the learned hidden state semantics to U and V gradually.
>     * We initially considered weight normalization to stabilize training but found it limited in this context, and layer normalization increases latency and slows down the training convergence. In the table below, we also provide a comparison with spectral normalization [1] (we turn to spectral normalization because we found spectral regularization is for generalizability rather than stability) and orthogonal regularization [2]. We hypothesize that these techniques are less effective because they are designed to ensure the backpropagated signal does not vanish or explode, and two of them constrain the weight much. However, the challenge of structured parameterization is not only about signal propagation but also the capacity bottleneck of learning a good representation, which the typical regularization techniques can't solve. For the constraints, the spectral normalization directly scales the trained weight by its largest singular value, which may hurt the performance here. The additional term in the orthogonal regularization also distorts the spectrum.
>
>       | LowRank (32% FFN params.)  | Perplexity |
>       |-|-|
>       | Baseline| 29.22|
>       |Self-guided training| **28.02** |
>       |Weight normalization | 29.12 |
>       |Spectral normalization | 31.93 |
>       |Orthogonal regularization | 29.18|
>
>     In the papers' appendix, we compared self-guided training with other techniques. We will also emphasize this discussion in the revision.
>
>    [1]. Spectral Normalization for Generative Adversarial Networks.
>
>    [2]. Can We Gain More from Orthogonality Regularizations in Training Deep CNNs?
>
> * **Q3** : It is also hard to identify whether the scaling law of BlockDense is necessarily better than the dense matrices in Figure 1a. ... Could you provide more quantitative analyses on the scaling laws (actual slope, confidence interval, etc.)?
>
>   **A3** : Thanks for your good suggestion. To make the illustration clearer, we applied the linear fit to the scaling results and provided them in the general response PDF. Based on Figure 1 in the PDF, we can calculated the slope and obtained the results outlined in the table below:
>    | Method | slope|
>    |-|-|
>    | Dense (optimal scaling law) | -0.3549 |
>    | 63% FFN params.  |  |
>    | LowRank | -0.3672 |
>    | BlockDense | -0.3673 |
>    | BlockShuffle | -0.3718 |
>    | 32% FFN params.  | |
>    |LowRank | -0.3852 |
>    |BlockDense| -0.3827 |
>    |BlockShuffle| -0.3881 |
>
>   The table shows that these structured parameterizations all have larger absolute slopes than the dense model at its optimal trade-off. This brings a smaller loss gap in the 1.3B model compared to the 110M models, indicating that structured parameterizations scale well or even better than the dense model. Additionally, their scaling curves for our structured parameterizations can be further optimized by finding a better balance between model size and training tokens.

---

> > ### Comment · Reviewer_5PEQ · 2024-08-11
> > **Response to the authors**
> >
> > Thanks for the rebuttal and the additional results! I have read the rebuttal and the replies to other reviewers.
> >
> > Q1 My concern is that since BlockDense is an intermediate between LowRank and BlockShuffle, and the performance & latency (even scaling law) results might not reach that of LowRank, the benefits of BlockDense are not sufficiently clear. Could you elaborate on the potential benefits that BlockDense would introduce in the vision tasks?
> >
> > Q2 The result of spectral normalization is great, but it would be more helpful to know whether the training loss spikes still exist for spectral normalization on low rank matrices (as the self-guided training is more oriented to training stability).
> >
> > Q3 Thanks for the new results! The figure 1 in the author rebuttal is much clearer now.
> >
> > At this moment, my Q3 and (half of) the Q2 are well addressed. I will raise my score to borderline accept.

---

> > > ### Author Response · Authors · 2024-08-13
> > > **Response to Reviewer 5PEQ**
> > >
> > > We would like to thank the reviewer for their swift response to our rebuttal. Below we further elaborate on our response to address the remaining concerns in Q1 and Q2  completely.
> > >
> > > * *For Q1*, we introduced BlockDense to cover a bigger parameterization space because it has consistent latency results with LowRank and achieves slightly better accuracy results than BlockShuffle within the FFN module and NLP tasks. Moreover, to show its benefits more clearly:
> > >
> > >   * Though the main focus of this paper is not to determine which method is the best, but rather to investigate their common problems, we find that this can be related to the data domain. Unlike in the FFN module of NLP tasks, we observe that block-diagonal matrices perform better in vision tasks. For example, the table below shows the ViT-small performance on the CIFAR-10 dataset. It can be seen that BlockDense surpasses LowRank by 0.6 points and it is much faster than BlockShuffle. We think this is because vision tasks tend to prefer locality due to the local correlations in the pixel space and block-diagonal matrices provide a more suitable inductive bias for that.
> > >   || Model Params (M) | CIFAR10 Acc |
> > >   |-|-|-|
> > >   | ViT (H=384)       | 21.3 | 92.5|
> > >   |LowRank      | 8.3| 89.6|
> > >   |BlockDense   | 8.3| 90.2 |
> > >   | BlockShuffle | 8.3 | 90.4 |
> > >
> > >   * The BlockDense parameterization can be seen as a generalization of the LowRank parameterization. For example, by simply setting B=1 for the first matrix, one can recover the low-rank parameterization. Expressing them in the same parameterization can allow us to explore hybrid and mixed structures more easily. As this paper’s aim is to explore their common problems, we leave exploring different hyper-parameters of BlockDense as the future work.
> > >
> > >
> > >
> > > * *For Q2*: We found that there are no training spikes with spectral normalization. However, it constrains the weights by scaling them via the spectral norm. Also, as we stated in our first response, the main problem is the capacity bottleneck of learning a good representation. The self-guided training allows the dense weights to transfer its learned hidden units' semantics to structured matrices, which suffer from symmetry problems during the feature specialization phase. As can be seen from Figure 3 in the submitted paper, it helps with both the slower convergence and loss spikes. This point, as well as non-constraints on weights, makes our technique different from and better than classical regularization methods.

---

> > > > ### Comment · Reviewer_5PEQ · 2024-08-14
> > > > **Response to the authors**
> > > >
> > > > Thanks for the swift reply.
> > > >
> > > > For Q1, I know that the purpose of this paper is to explore the common issues of using structured matrices training in FFN. My concern is that because BlockDense is an intermediate between low rank and BlockShuffle and BlockDense's performance is fully dominated by low rank matrices *without the results of vision tasks*, **the value of even discussing BlockDense in the context of LLM training is not clear**. So I would suggest the authors to add the discussion of vision tasks to the paper.
> > > >
> > > > For Q2, one additional study that would further strengthen Q2 is to give a more concrete empirical evidence on the causal relationship between capacity bottleneck and training loss spikes. Currently such evidence is still missing in the paper.
> > > >
> > > > I will maintain my score at this moment.

---

> ### Author Response · Authors · 2024-08-14
> **Response to Reviewer 5PEQ**
>
> Thanks for your quick reply and suggestions.
>
> * For Q1, yes, we’ll add the table of vision results and also discussion into the paper to help readers build a better understanding of different parameterizations. For the FFN module in NLP tasks, the BlockDense still needs to be further explored as we stated during the rebuttal and in our paper. As a result, we didn’t claim the proposed BlockDense method as the main contribution of this paper. Also, we tend to keep it in the paper to cover a bigger parameterization space, and it’s a generalizable version of LowRank. Given its good performance, we think keeping it can be more informative at least as an ablation to LowRank parameterization.
>
> * For Q2, the training instability arises from the deep linear form U(Vx), which introduces additional symmetries and thus a more complex loss landscape to optimize over [1, 2]. To illustrate the relationship between loss spikes and capacity loss, we show in the table below that models with high rank suffering from severe loss spikes can even perform worse than models with very low rank. This table represents our early experiments where we first observed loss spikes with LowRank on the CIFAR-10 dataset while sweeping learning rates. We highlight poor results in bold. For higher ranks, we found it easier to experience severe loss spikes at very large learning rates. For instance, the Rank 256 model shows an accuracy of 87.50, which is worse than the worst result for Rank 8. The Rank 8 model is 32 times smaller than Rank 256 and has poor results due to slower convergence. However, the performance can still be better than the Rank 256 with severe loss spikes.
>
>     | lr        | 1.0e-4 | 2.5e-4 | 5.0e-4 | 7.5e-4 |
>     |----------------|------------|------------|------------|------------|
>     | Rank 4    | **87.49**  | **89.77**  | 91.22      | 90.02      |
>     | Rank 8     | **88.42**  | **90.47**  | 90.89      | 90.55      |
>     | Rank 16    | **88.81**  | **91.07**  | 92.31      | 92.21      |
>     | Rank 128   | 91.28      | 93.14      | 93.73      | **89.10**  |
>     | Rank 256   | 91.95      | 93.60      | 93.16      | **87.50**  |
>     | Dense 384| 91.86      | 93.67      | 93.66      | 93.36      |
>
>    This is a very good question that we were also curious about investigating, especially early in the project. The results we provided in the table above are very preliminary results that we obtained on CIFAR10 using VIT architecture. Let us note that other hyperparameters of this table and, thus the dense results are not the same as that in our last response because they’re very early results. As we discovered that the self-guided training addresses the optimization challenge without tuning hyper-parameters, we decided not to include these results in the submitted version of our paper. However, we will include a more elaborate discussion about the loss spikes and training instability in the camera-ready version of the paper, with the detailed results presented in the appendix.
>
>   [1]. Exact solutions to the nonlinear dynamics of learning in deep linear neural networks
>
>   [2]. Neural networks and principal component analysis: Learning from examples without local minima.

---

### Official Review · Reviewer_7Gik · 2024-07-12

**Soundness:** 3
**Presentation:** 3
**Contribution:** 2
**Rating:** 7
**Confidence:** 3

**Summary:**

The paper studies efficient Transformer variants. Unlike most existing works on efficient attention, this work proposes methods to enhance the efficiency by focusing on feedforward networks (FFNs). It explores several efficient linear layer designs, and proposes techniques to address the training issues and decoding efficiency for practice. The experiments show that these structured FFNs not only reduce computational costs but also show promising scaling behavior on language modeling.

**Strengths:**

- Paper is clearly written and well motivated.

- While there has been voluminous literature on efficient attention using structured matrices, making FFNs efficient using structured matrices is new to the community.

- This paper discuss interesting techniques to address optimization difficulty in training Transformers with structured matrices in FFNs.

- The experiments covers models with different sizes and demonstrate the scaling behavior. The experiments also clearly demonstrate the efficiency gains in practice.

**Weaknesses:**

- The experimental study can be made more solid if the authors can additionally provide model quality analysis on downstream tasks (e.g., SuperGLUE) and the finetuning regime.

- The second part of the related work section could be enhanced by including additional studies on structured matrices in Transformers, emphasizing that the majority of existing efforts focus primarily on the attention module. To name a few:

  - Lee-Thorp, James, et al. "Fnet: Mixing tokens with fourier transforms." arXiv preprint arXiv:2105.03824 (2021).

  - Luo, Shengjie, et al. "Stable, fast and accurate: Kernelized attention with relative positional encoding." Advances in Neural Information Processing Systems 34 (2021): 22795-22807.

  - Choromanski, Krzysztof, et al. "From block-Toeplitz matrices to differential equations on graphs: towards a general theory for scalable masked Transformers." International Conference on Machine Learning. PMLR, 2022.

- Minor issues.
  - Lines 4 & 71: transformer $\to$ Transformer
  - Line 168: alpha $\to$ $\alpha$

**Questions:**

Is there any special technique to initialize the matrices U and V? What is the scale/variance of the initialization?

**Limitations:**

The authors discuss the limitation of this paper in Sec. 5. I believe that another limitation is that the paper lacks evaluations on downstream tasks.

---

> ### Author Rebuttal · Authors · 2024-08-07
>
> We would like to thank the reviewer for the valuable suggestions and careful reading of this paper. We will fix the minor issues in the revision and list the detailed responses to the questions below. Hope our reply can address the concerns.
>
> * **Q1**: The experimental study can be made more solid if the authors can additionally provide model quality analysis on downstream tasks (e.g., SuperGLUE) and the finetuning regime.
>
>     **A1**: Thank you for your valuable suggestion. We have added the results of downstream performance in Table 1 and Table 2 in the general response PDF, as well as the fine-tuning performance in the table below.
>     * We use the lm-evaluation-harness repository for downstream tasks like PIQA and HellaSwag. To achieve good downstream performance, we train small-sized Transformers on 100B tokens. Table 1 in the general response PDF shows the consistent performance of these zero-shot tasks with the validation perplexity of training. All structured parameterizations with 32% FFN parameters have results close to the dense models (e.g., a 0.9-1.4-point averaged accuracy decrease). Additionally, we also provide the results of Transformer-XL trained based on the optimal scaling law in Table 2 in the response PDF, showing good results for structured matrices and consistent improvement with self-guided training.
>     * We applied fine-tuning to our trained models (small-sized Transformers on 100B tokens) using the famous transformer repository. As GLUE is better supported in this codebase than SuperGLUE, we evaluated two tasks : QQP and SST-2 in the GLUE benchmark within this short rebuttal period. As can be seen below, the structured parameterizations show very comparable accuracy performance to the dense model (e.g., 0.3 lower accuracy of BlockDense on QQP task) with only 32% parameters of the FFN module.
>
>        |     Method              | Validation PPL | QQP (acc) | SST-2 (acc) |
>        |----------------------|----------------|-----------|-------------|
>        | Transformer-s (110M) | 16.68 | 90.0      | 92.0  |
>        | 32% FFN params.      |   |  ||
>        |     LowRank          | 18.65 | 89.6 | **92.2** |
>        |     BlockDense       | **18.61**      | **89.7**  | 91.7  |
>        |     BlockShuffle     | 18.94          | 89.2      | 91.5 |
>
>       To conclude, the downstream and fine-tuning performance consistently shows the strong potential of these structured matrices.
>
> * **Q2**: The second part of the related work section could be enhanced by including additional studies on structured matrices in Transformers, emphasizing that the majority of existing efforts focus primarily on the attention module.
>
>   **A2**: Thanks for the good suggestion. We'll add these papers to the related works. Yes, they are all related and very interesting papers that apply structured matrices, like Toeplitz, over the sequence dimension, thus making the attention module very efficient. In contrast, in our paper, we investigated structured parameterizations in the FFN module and found that Toeplitz is not very suitable for mixing information in the hidden states dimension for NLP tasks. We also focus on their scaling, optimization, and efficiency aspects in modern LLM architecture training.
>
> * **Q3**: Is there any special technique to initialize the matrices U and V? What is the scale/variance of the initialization?
>
>   **A3** : We put the discussion about initialization in Section B.1 of the submission, where we mentioned that we used spectral initialization for LowRank and orthogonal initialization for BlockDense and BlockShuffle. Specifically, we use random Gaussian initialization with a variance of 0.02, similar to GPT-2, to initialize the original weight W. Then, as suggested by [1], U and V in LowRank parameterization are initialized by using the SVD decomposition of W. For the other two methods, we apply orthogonal initialization as recommended in the paper on deep linear networks [2]. Experiments in Section B.1 on the small dataset WikiText-103 further validated our choice.
>
>     [1]. Initialization and regularization of factorized neural layers. ICLR'21
>
>     [2]. Exact solutions to the nonlinear dynamics of learning in deep linear neural networks. Andrew M. Saxe, et al. 2014.

---

> > ### Comment · Reviewer_7Gik · 2024-08-09
> > **Thank you for the rebuttal**
> >
> > I thank the authors for the rebuttal. All my concerns are addressed.
> >
> > For GLUE I would recommend finetuning on the mixture of all the data and evaluate on each task individually - that can be both efficient and informative. That being said, the reported results in the rebuttal look promising to me.

---

> > > ### Author Response · Authors · 2024-08-13
> > > **Response to Reviewer 7Gik**
> > >
> > > Thank you for your quick reply and suggestion. Based on your suggestion, we trained the model with a binary classification head on a mixture of the GLUE benchmark, excluding MNLI and STS-B, since they are not binary classification tasks and thus are difficult to mix with the others.
> > >
> > > We trained the models for 6 epochs, using a batch size of 128 and a swept learning rate. From the table below, we observe that structured parameterizations exhibit performance close to that of the dense model (e.g., a 0.64 accuracy loss for BlockDense).
> > > Meanwhile, we also noticed that the performance on very small datasets, including CoLA, MRPC, and RTE, is not very stable.
> > >
> > > This might be fixed with further hyperparameter search with more expensive runs, because we believe that the smaller datasets tend to be more sensitive to the hyperparameters. Moveover, we didn’t consider the weight of each dataset when mixing them, which may also not be preferable for the smaller datasets.
> > >
> > > || Validation PPL | CoLA (matt.) | SST-2 (acc) | MRPC (acc) | QQP (acc) | QNLI (acc) | RTE (acc) | Avg.  |
> > > |--|-|-|-|-|-|-|-|-|
> > > | Dense           | 16.68          | 45.32        | 90.48       | 80.15      | 89.71     | 86.36      | 64.98     | 76.17 |
> > > | 32% FFN Params. |                |              |             |            |           |            |           |       |
> > > | LowRank         | 18.65          | 43.01        | 90.13       | 80.88      | 89.50     | 86.09      | 63.18     | 75.47 |
> > > | BlockDense      | 18.61          | 42.16        | 90.14       | 78.67      | 89.82     | 86.34      | 66.06     | 75.53 |
> > > | BlockShuffle    | 18.94          | 44.51        | 89.79       | 79.90      | 89.36     | 85.56      | 61.73     | 75.14 |

---

> > > > ### Comment · Reviewer_7Gik · 2024-08-13
> > > >
> > > > Thank you for the update. I think the additional experiments in the rebuttal make the paper more solid. I raise my rating from 6 to 7 to support the paper to be accepted.

---

### Author Rebuttal · Authors · 2024-08-07

We would like to thank all the reviewers for their constructive feedback. Before replying to the comments one by one, we would like to highlight our contributions and clarify common questions in this general response:

In this paper, we investigate the performance of three structured parameterizations within the FFN modules in modern LLM architecture training, focusing on efficiency, optimization, and scaling aspects. In detail,

  * We conducted a comprehensive efficiency study in various scenarios, using our proposed pre-merge technique, to show their good latency performance.
  * We identified the optimization challenges of the structured parameterizations and proposed an effective method called self-guided training to boost performance for all of them.
  * We showed that these structured matrices have steeper scaling curves and can utilize training FLOPs more effectively than the dense model. To further validate this point, we trained a wide and structured network of medium size to have fewer parameters and lower perplexity.

For common concerns, we added the downstream performance in Table 2 and Table 3 in the attached PDF, showing consistent performance with validation perplexity of pre-training, and close results to the dense model. Additionally, we applied a linear fit to the scaling curves to make the illustration clearer in Figure 1 of the attached PDF.

---

### Decision · Program_Chairs · 2024-09-25

**Decision:**

Accept (poster)

**Comment:**

The paper studies structured parameterizations (LowRank, BlockShuffle, and BlockDense) for the FFN or MLP layers to improve the efficiency of the transformers. Unlike many existing works that focus on fine-tuning, this work exploits these structured weight matrices during the pretraining stage. To address the training challenges associated with structured parameterizations, the authors introduce self-guided training, which uses a dense matrix as a residual component that is gradually annealed. Experiments show that the proposed method achieves a 2.5x speed-up with only a 0.4 PPL increase under the same training FLOPs.

While the use of structured weight matrices is not new and has been widely studied before for other architectures (such as ResNet), their exploration in the context of LLMs, particularly for pretraining, is timely and useful for improving the efficiency of LLMs. Reviewers pointed out several issues in the experiments, most of which were addressed by additional experiments in the rebuttal. The authors could include these in the final version to make the paper more robust.